# Producing Metal Powder from Machining Chips Using Ball Milling Process: A Review

**DOI:** 10.3390/ma16134635

**Published:** 2023-06-27

**Authors:** Leong Kean Wei, Shayfull Zamree Abd Rahim, Mohd Mustafa Al Bakri Abdullah, Allice Tan Mun Yin, Mohd Fathullah Ghazali, Mohd Firdaus Omar, Ovidiu Nemeș, Andrei Victor Sandu, Petrica Vizureanu, Abdellah El-hadj Abdellah

**Affiliations:** 1Faculty of Mechanical Engineering & Technology, Universiti Malaysia Perlis, Arau 02600, Malaysia; kwleong@unimap.edu.my (L.K.W.); allicetanyin@gmail.com (A.T.M.Y.); fathullah@unimap.edu.my (M.F.G.); 2Center of Excellence Geopolymer and Green Technology (CEGeoGTech), Universiti Malaysia Perlis, Kangar 01000, Malaysia; mustafa_albakri@unimap.edu.my (M.M.A.B.A.); firdausomar@unimap.edu.my (M.F.O.); 3Faculty of Chemical Engineering & Technology, Universiti Malaysia Perlis, Kangar 01000, Malaysia; 4Department of Environmental Engineering and Sustainable Development Entrepreneurship, Faculty of Materials and Environmental Engineering, Technical University of Cluj-Napoca, B-dul Muncii 103-105, 400641 Cluj-Napoca, Romania; 5Faculty of Materials Science and Engineering, Gheorghe Asachi Technical University of Iasi, Blvd. D. Mangeron 71, 700050 Iasi, Romania; sav@tuiasi.ro (A.V.S.); peviz@tuiasi.ro (P.V.); 6Romanian Inventors Forum, Str. Sf. P. Movila 3, 700089 Iasi, Romania; 7Technical Sciences Academy of Romania, Dacia Blvd 26, 030167 Bucharest, Romania; 8Laboratory of Mechanics, Physics and Mathematical Modelling (LMP2M), University of Medea, Medea 26000, Algeria; lmp2m_cum@yahoo.fr

**Keywords:** ball milling, recycling metals, milling parameters, mechanical properties, rapid tooling, mechanical alloying, mould additive manufacturing

## Abstract

In the pursuit of achieving zero emissions, exploring the concept of recycling metal waste from industries and workshops (i.e., waste-free) is essential. This is because metal recycling not only helps conserve natural resources but also requires less energy as compared to the production of new products from virgin raw materials. The use of metal scrap in rapid tooling (RT) for injection molding is an interesting and viable approach. Recycling methods enable the recovery of valuable metal powders from various sources, such as electronic, industrial, and automobile scrap. Mechanical alloying is a potential opportunity for sustainable powder production as it has the capability to convert various starting materials with different initial sizes into powder particles through the ball milling process. Nevertheless, parameter factors, such as the type of ball milling, ball-to-powder ratio (BPR), rotation speed, grinding period, size and shape of the milling media, and process control agent (PCA), can influence the quality and characteristics of the metal powders produced. Despite potential drawbacks and environmental impacts, this process can still be a valuable method for recycling metals into powders. Further research is required to optimize the process. Furthermore, ball milling has been widely used in various industries, including recycling and metal mold production, to improve product properties in an environmentally friendly way. This review found that ball milling is the best tool for reducing the particle size of recycled metal chips and creating new metal powders to enhance mechanical properties and novelty for mold additive manufacturing (MAM) applications. Therefore, it is necessary to conduct further research on various parameters associated with ball milling to optimize the process of converting recycled copper chips into powder. This research will assist in attaining the highest level of efficiency and effectiveness in particle size reduction and powder quality. Lastly, this review also presents potential avenues for future research by exploring the application of RT in the ball milling technique.

## 1. Introduction

Manufacturing industries have witnessed increasingly dynamic marketplaces and rising levels of competition throughout time. While mold additive manufacturing (MAM) may be found in the prominent scientific literature, MAM has yet to be commercialized [1,2,3,4]. Furthermore, many businesses are constantly seeking cutting-edge technologies to satisfy growing demand and lower prices, and enable the production of products in small quantities without compromising quality. Additionally, businesses aim to achieve sustainable development goals [5,6,7]. The advancement of technology and the increase in population have led humans towards consumerism, resulting in exponential industrial expansion with environmental consequences such as mineral expending, energy production, and waste creation, as witnessed in the steel industry. As a result, the concept of the circular economy is being discussed in terms of recycling or adding value to scrap materials by developing new applications [8]. Metal chips are generated during machining operations and are often viewed as waste. However, due to their high metal content, these chips can be reused in different applications, resulting in both environmental and economic benefits.

Numerous studies have assessed the reuse of metal chips in various applications, for example, the replacement of virgin metal by recycled chips in mold application. These studies have demonstrated that using recycled metal chips in mold production can save costs, reduce environmental impact, and improve product quality. Figure 1 illustrates the various types of machining chips produced by different manufacturing processes, including turning, milling, laser cutting, facing, stamping, drilling, and grinding [9,10,11]. The morphologies of metal chips are determined by the cutting parameters and the materials used for both the workpiece and the cutting tools [12]. The chips produced during the processes of grinding and laser cutting can be directly recycled due to their small size, eliminating the need for pre-processing. However, the presence of wheel impurities and the occurrence of high-temperature oxidation restrict the utilization of these machining chips for recycling purposes. As a result, the majority of the chips available for recycling are sourced from other machining operations such as turning, drilling, end-milling, and shaping.

In addition, metal scraps are generated as large waste in the machining industry, which is used for powder production as part of goal 12 of the 2030 Agenda (Global Transformation: The Sustainable Development of 2030 Agenda) [26]. By recycling metal scraps, the industry can contribute to reducing waste and minimizing the extraction of new raw materials.

There are several advantages to recycling metal waste:i.Resource conservation: Recycling metal waste conserves natural resources, as it reduces the need for mining and processing new raw materials [27,28,29]. This helps to preserve the environment and reduces the energy required for manufacturing.ii.Energy conservation: Recycling metal waste requires less energy than extracting and refining raw materials, as the metal has already been processed [27,30,31]. This means less energy is required to manufacture new products from recycled metal.iii.Reduce landfill waste: Recycling metal waste reduces the amount of waste that goes to landfills, as it can be used to create new products [32,33,34].iv.Reduce greenhouse gas emissions: Recycling metal waste results in fewer greenhouse gas emissions, as the energy required for recycling is generally lower than the energy required for manufacturing new metal products [35,36].v.Economic benefits: Recycling metal waste can create jobs and economic opportunities in the recycling industry. It can also reduce the costs of manufacturing new metal products, as recycled metal is often cheaper than newly produced metal [37,38,39].

Metal fragments can be used to produce new metal products and can be recycled to reduce waste and conserve natural resources [40,41,42]. Machining chips are estimated to account for a significant portion (13.7% aluminum and 14.6% steel) of the trash created by all manufacturing processes globally [43]. This includes metal waste generated from machining processes such as mild steel, aluminum, copper, and brass [44], as illustrated in Figure 2. One potential application for metal chips is in the production of molds. On the whole, metal scrap recycling and reuse can reduce greenhouse gas emissions and energy consumption, making it more sustainable [45,46,47]. Several endeavors have been made to recycle metal chips through compression and extrusion processes. However, this consolidation technique is limited to parts with a consistent cross-section [20]. To manufacture intricate components from metal scrap chips, a proposed approach involves mechanically milling the chips into a powder suitable for application in MAM. Therefore, ball milling offers advantages such as efficient size reduction, versatility, mixing capabilities, scalability, energy efficiency, control over particle size, ease of operation and maintenance, and wide-ranging applications.

Among various metal matrix composites (MMCs), the 7xxx series, which consists of minor alloying elements such as Zn, Mg, and Cu, have garnered more attention compared to conventional metal-based alloys for use as matrix materials. This is due to their high specific mechanical properties, lightweight nature, and easy availability [49,50]. Moreover, copper holds significant importance in the manufacturing industry as it is extensively utilized for the production of a wide variety of products. However, the literature on recycling highlights that the efficiency of remelting-based recycling processes for metallic scraps, including copper, is relatively low [51]. Therefore, an alternative approach to recycling copper chips without the need for a melting process can offer several advantages. This includes the potential for increased economic profit and a reduction in the environmental impact associated with the related industry [52].

The technique of mechanical alloying (MA) was originally developed in 1965 to meet the industrial demand for oxide-dispersion-strengthened (ODS) nickel-based superalloys [53]. Since then, MA has undergone extensive research and has found applications in diverse fields, resulting in the exploration of various new applications for mechanically alloyed materials, as depicted in Figure 3. In recent times, mechanical alloying (MA) has gained popularity for its ability to disperse various types of carbide (TiC [54], B_4_C [54], SiC [54]), oxide (Al_2_O_3_ [54], Y_2_O_3_ [55], TiO_2_ [56]), and carbon-based (CNT [57], SWCNT [57], FWCNT [57], MWCNT [57]) reinforcements into Al-based composites. MA, which is a straightforward mechanical milling technique, offers the advantage of producing uniform nanocrystalline materials [58] and nanocomposites through extensive plastic deformation and fracture mechanisms caused by the impact of small and rigid balls [59,60]. Additionally, MA effectively breaks down the flexible and entangled CNTs, resulting in a homogeneous dispersion of CNTs within the matrix [60]. Nevertheless, controlling the structural and morphological evolution of CNTs in the MA process can be challenging due to the optimization of numerous ball milling parameters [61].

Mechanical alloying as a method of solid-state particle processing that involves mechanical welding, breaking, and rewelding particle powders repeatedly in a high-energy ball mill [53,62,63]. It is among the most effective nanostructures top-down preparation procedures for nanocrystalline [64], nanoparticles [65], and nanocomposite materials [66]. Due to its widespread usage and significance in materials science, metallurgy, and nanotechnology literature, the term “mechanical alloying” (MA) is commonly employed to describe high-energy ball milling, making it the most prevalent top-down technique [67]. The top-down method was deemed a promising option to easily create nanomaterials in large numbers by Vu et al. [68]. In this scenario, copper may have its surface modified by ball milling, which is an easy and practical procedure. The nanoscale is essential for low-cost synthesis methods. In order to establish mechanical milling as a feasible choice for mold additive manufacturing (MAM) feedstock, it is crucial to comprehend the correlation between the processing parameters and the morphology, particle size, and chemical composition of the powder [69,70,71]. Likewise, examining the novelty of refinement recycled copper chips processing settings and circumstances is critical when using mechanical milling methods such as the planetary ball mill process. Fundamentals of this ball milling strategy are discussed in the following section.

## 2. Ball Milling

The first ball mill was created by the German physicist and chemist Friedrich Fischer in the late 19th century [72]. Since then, ball milling’s ease of use, adaptability, and scalability have made it a staple in the fields of materials science and engineering. Ball milling is a common method for decreasing particle size and increasing material reactivity [73,74]. The process uses a ball mill, a cylindrical chamber with a grinding medium such as balls that rotates on its axis to achieve the desired fineness of the final product [9,75]. The grinding media crash into the particles, reducing their size and creating a more consistent blend [76,77]. Ball mills find applications across many disciplines, from materials research to chemical analysis and even biological research [78,79,80]. Ball milling is a process that utilizes mechanical forces, such as those generated by the rotating jar of a ball mill, to break up the material into smaller pieces and mix them together [81]. Ball milling is a versatile technology that has many uses [82]. It may be used to make high-quality powders, disperse nanoparticles, and even synthesize complicated materials [78,79,82,83].

Firstly, the strategy of ball milling is a mechanical approach to decreasing the size of particles of materials, such as powders and suspensions [53,84]. This strategy is notable since the usage of these materials appears to be sustainable [26,85]. A ball mill is additionally utilized to grind and combine materials, commonly with spherical or cylindrical-shaped grinding media such as balls or rods [86]. Numerous researchers have effectively used ball milling to produce nanostructures of various materials or to investigate structural changes in materials during ball milling [87,88,89,90,91]. Moreover, planetary ball mills are the most popular mills used in MA and MM processes for synthesizing almost all types of metastable and composite materials. In mills of this type, the milling media possess significantly high energy due to the movement of the milling stock and balls against the inner wall of the vial (milling bowl or jar). As a result, the effective centrifugal force can reach up to twenty times the acceleration due to gravity [92]. The ball mill typically consists of a cylindrical shell that spins around its axis and the grinding medium is placed inside the shell [93]. As the shell rotates, the grinding medium is lifted and dropped onto the material being processed, resulting in particle size reduction, as shown in Figure 4.

The findings reveal that, under the influence of impact forces, the original particles undergo welding, forming a sheet. Subsequently, the sheet undergoes thinning and enlargement due to the squeezing effect. With an extended duration of ball milling, the accumulation of stress concentration leads to the emergence of cracks in the sheet. Consequently, the sheet fractures into smaller fragments. As ball milling continues for a longer period, these fragments further break down into significantly smaller pieces. Eventually, the particles reach a dynamic balance stage where their size stabilizes [67,68]. The ball milling method may be used to generate powders with a wide variety of particle sizes and is often employed in manufacturing metal powders, ceramic powders, and other materials. This method may also be utilized for mixing and blending materials and for chemical processes, such as solid-state reactions and mechanochemical reactions [94,95]. The conversion of metal chips into powder requires intensive grinding. The chips can be milled for crushing under conventional conditions, with and without a cryogenic environment [96], to optimize the particle sizes and other qualities of the finished products [97].

Furthermore, ball milling may be used to manufacture metal powders from waste metals. The process of ball milling involves the grinding and mixing of metal particles within a ball mill, which is a type of equipment used for grinding, with the aim of producing a fine powder [98]. This method has been widely employed in the manufacturing of metal powders from diverse sources, including metals, alloys, and intermetallic compounds. With a ball milling technique, it is feasible to produce machining chips into metal powder. However, different cutting conditions result in different shapes and sizes of chips. For instance, the Hardox 500 steel chips morphology reflects the details of the CNC milling cutting mechanism and, under different cooling and lubricating conditions, presents regular serrations along with the “c” and “s” shapes of chips [99]. The various shapes/sizes and greasy or oily metal chips should be considered separately before the ball mill process. Therefore, chips are currently cleaned by decanting, centrifugation, briquetting, and acetone washing [100]. In addition, the chips were washed by soaking them in room-temperature acetone and agitating them for 15 min [101]. Cleaning would be unnecessary if the chips were made using a dry machining operation. The machining chips are put into a ball mill together with the grinding medium in this process after being cleaned and dried to eliminate any impurities [96,102,103]. The chips are subsequently ground into a fine powder while the ball mill rotates using the grinding medium [9,104,105]. In order to achieve the ideal particle size and other attributes of the finished copper powder product, it is possible to modify the grinding medium size and shape, milling duration, and milling intensities [105,106].

On the contrary, there are several potential disadvantages associated with ball milling. These include the possibility of contamination, formation of nanomaterials with irregular shapes, noise generation, and long milling and cleaning times [76]. Different types of equipment are also available depending on the material to be treated, which have been described in detail in a recent review by Gorrasi and Sorrentino [107]. One drawback of ball milling is that it is a time-consuming process, especially when fine milling or high-quality powder production is desired. Achieving the desired particle size and distribution often requires extended milling times, leading to longer processing durations. Additionally, the friction and collisions between the balls and the material during milling can generate significant heat, which may have an impact on the stability and properties of the milled material, particularly for heat-sensitive materials. Hence, the selection of parameters in the ball milling process plays a vital role in determining the ultimate properties of the product [108,109]. It can significantly contribute to improving the performance of the ball mill process, especially in terms of refining the powder within a minimum ball milling time. Therefore, the following section provides a comprehensive discussion of the various parameters involved.

### 2.1. Important Processing Parameters in the Ball Mill Process

Ball milling is a widely utilized technique in the field of materials science and engineering for the synthesis of various advanced materials [109,110,111]. The process involves the mechanical activation of powders through the repeated collision of milling balls with the powder particles, reducing particle size and creating new interfaces [112,113]. While the ball milling process is relatively simple, there are several critical processing parameters that can significantly impact the qualities and properties of the final product [114,115,116]. In this context, this article discusses some of the important processing parameters in the ball milling process.

#### 2.1.1. Ball-to-Powder Ratio (BPR)

During high-energy planetary ball milling, numerous collisions occur between the balls and the powder, as well as between the balls and the jar. These collisions are more intense and frequent, leading to more severe impacts. As a result, the particles undergo deformation and flattening, resulting in flattened particle morphology. Additionally, the repeated impacts and shearing forces contribute to the reduction in particle size, leading to smaller particles [103,104]. The microstructure of Tungsten was improved with the use of the BPR model developed by Wu et al. [117]. The authors conducted experiments by varying the ball-to-powder ratio (BPR) and comparing the results using different structural tools. It was observed that as the BPR increases, the frequency of collisions and the intensity of stress caused by the velocity of the milling ball also increase [118]. This leads to a reduction in particle size, eventually reaching the nanoscale, as depicted in Figure 5.

#### 2.1.2. Milling Time

One of the critical processing parameters in ball milling is the milling time, which determines the extent of powder particle size reduction and the level of homogenization achieved in the material. Generally, longer milling times lead to smaller particle sizes, but there is a limit beyond which the particles can become overly fine and unstable. For instance, Eskandarany et al. [119] investigated the impact of milling time on the particle size distribution and morphology of MgH_2_ powders and reported that prolonged milling led to the formation of agglomerated particles, which adversely affected the hydrogen desorption kinetics. The milling duration affects the degree of mechanical activation and particle size and distribution, as well as the final structure and qualities of the product. Longer milling times typically result in smaller particle sizes, higher surface areas, and increased defects [120], with no phase change occurring during the milling process [121]. Similarly, Salur et al. [55] conducted a study to examine the morphological transformation of AA7075-0.5 wt% Y_2_O_3_ (Yttrium oxide) composites during planetary ball milling for up to 10 h. According to Figure 6, the average particle size reached 16 µm after 10 h of milling. As the milling time was prolonged, the specific surface area of the milled composites gradually increased owing to the reduction in particle size.

Furthermore, a study conducted by Kishore et al. [122] investigated the effect of milling duration on the particle size of titanium alloy (Ti-6Al-4V) powders with 0.3 wt.% Ca (calcium). The study reported that the particle size of 175 to 250 μm (2 h), 100 to 400 μm (4 h), and 75–100 μm (6 h) with a ball-to-powder weight ratio of 10:1 at a milling speed of 350 rpm. The technique mentioned can also be employed to assess changes in crystallite size and lattice strains caused by powder ball milling at various durations [123], as depicted in Figure 7. Furthermore, an increase in mechanical strength and flexural strength can be observed with an increase in milling time. In contrast, a shorter milling time could result in agglomeration, but as the time is extended, the probability of agglomeration weakens, leading to improved mechanical properties [124]. Dogan et al. [125] conducted a comprehensive investigation to assess the impact of milling time on the morphologies of aluminum alloy (AA7075) powder for refining small particles. After 1 h of milling, the average particle size was observed to be approximately 30 μm, and this size remained consistent after 2 h of milling. Surprisingly, when the milling time was extended to 4 h, the average particle size increased to around 31 μm, contrary to the expected decrease caused by particle fragmentation. This increase was attributed to the presence of numerous agglomerated particles. However, upon further milling for 8 h, a significant reduction in the average particle size to 11 μm was achieved. Additionally, the microstructures and mechanical properties of hot-pressed carbon nanotubes/aluminum (CNTs/Al7075) nanocomposites were assessed through optical microscopy, hardness measurements, and relative density analysis. Moreover, the dominant hardening mechanism in the CNTs-Al7075 hybrid system was believed to be the Orowan mechanism. A significant increase in Brinell hardness was observed, with the composite produced from 8 h of milling showing a remarkable increment from 91 to 237 HB, representing an approximately 160% improvement.

#### 2.1.3. Milling Speed

Another important parameter in ball milling is the milling speed, which determines the frequency and intensity of the milling ball collisions. Xu et al. [126] studied the deformation and dispersion mechanisms of aluminum (Al)–carbon nanotube (CNT) powder mixtures were found to be influenced by the milling speed. When the ball mill operated at a low speed of 135 rpm, the Al powders underwent gradual flattening, while the CNTs were uniformly dispersed onto the Al flakes with minimal damage. However, when the ball mill operated at a high speed of 270 rpm, the Al powders experienced rapid flattening and subsequently cold welded into particles. On the contrary, as depicted in Figure 8, the carbon nanotubes (CNTs) maintained their clustered form and experienced notable damage when subjected to high-speed milling conditions. Zhou et al. [127] examined how milling speed and milling media affected the production of magnetic covalent organic frameworks (COFs). The grinding speed describes the rotation speed of the grinding container, which affects the kinetic energy of the grinding balls and the effect energy of the powder [128]. Hussain et al. [129] extensively investigated the impact of different ball milling speeds while keeping the milling time constant at 30 h on the particle size of Ni70Mn30 samples. They observed a gradual decrease in particle size when the samples were milled at 200 and 300 rpm, attributing it to the collisions between the balls and alloy particles during milling. However, at a milling speed of 400 rpm, a noticeable increase in particle size was observed due to the simultaneous occurrence of cold welding and micro forging processes. When the milling speed was further increased to 500 and 600 rpm, there was only a slight reduction in grain size, likely resulting from particle fracturing caused by the higher speeds. The variation in average grain size with increasing milling speed is clearly depicted in Figure 9. Higher milling speeds lead to higher impact energies, resulting in smaller particle sizes and increased defects. However, very high speeds will lead to excessive heating and thermally induced reactions [129]. Relatively, to obtain a homogeneous distribution and dispersion of nanofillers within the matrix, relatively high grinding energy is required. Wet milling is less effective due to the presence of a solvent, while high-energy milling during dry milling can reduce the crystal size [130,131].

#### 2.1.4. Type of Milling Container

The milling container utilized can have a significant impact on milling efficiency, contamination levels, and the properties of the final product. Various factors, such as the material, shape, and size of the container, can influence the milling process. The ductility of the container material plays a crucial role in determining its resistance to impacts. A ductile material has the ability to absorb a significant amount of impact energy, thereby preventing alterations in the particle shape resulting from subsequent impacts [132]. Liu et al. [133] reported that the pan milling mechanochemical technique is superior to high-energy ball milling for the synthesis of stable anatase titanium dioxide/graphene nanosheets (TiO_2_/GNS). The custom-made pan-milling equipment comprises a mobile and stationary pan that can generate substantial compressive forces vertically and shearing forces parallelly, resembling the action of 3D scissors. During the ball-milling process, the metastable anatase TiO_2_ undergoes crystal structure disintegration due to the intense impact and friction forces. This is accompanied by a rise in local temperature (ranging from 160 to 200 °C) [121,122]. Consequently, the TiO_2_ particles gradually aggregate and form lumpy structures. However, important tests such as the compressive strength evaluation and visual observation of the resulting powders from both ball milling techniques were not conducted. Another limitation of the study is the exclusive focus on metal powders as the experimental samples. Therefore, future research should consider exploring the use of recycled copper chips to evaluate the ionic conductivity of composite electrodes.

#### 2.1.5. Milling Atmosphere

The milling atmosphere, which refers to the gas or vapor inside the milling container, plays a crucial role in various milling factors. It can influence chemical reactions, oxidation/reduction processes, and contamination levels during milling. Another experiment employed two different gaseous atmospheres, nitrogen and ammonia, to examine the nitridation of Fe powder at three different temperatures, including liquid nitrogen temperature, ambient temperature, and 200 °C [134]. Different atmospheres, such as vacuum, inert gas, or reactive gas, can have different effects on the milling process and the final product properties [135,136,137]. Umeda et al. [138] conducted a study to establish a solid-state recycling process specifically for Ti-6%Al-4%V (Ti-64) alloy waste. The aim of the study was to effectively produce Ti powder that could be used as raw starting materials for powder metallurgy (PM) components. The brittle titanium hydride (TiH_2_) powders’ compounds formation in Ti chips via heat treatment in hydrogen (H_2_) and (Ar) argon mixed gas atmosphere significantly improved their ball milling ability. During the milling process, the 10 g of each Ti-64 chip (after hydration treatment) were crushed and ground using of ZrO_2_ media balls (120 g) with a diameter of 10 mm. The jar rotated at a speed of 200 rpm, and the milling time ranged in 10, 30, and 60 min with morphology changes as shown in Figure 10. The powders obtained from milling without hydration treatment and those from milling with hydration treatment at 673 K exhibited similar size distributions, with a median particle diameter of 460 μm. However, when hydration treatment was performed at higher temperatures of 873 K and 1073 K, the median particle diameters were measured as 119.1 μm and 189.2 μm, respectively. The findings of this study prove that recycled Ti chips via heat treatment in hydrogen atmosphere (783 K) can produced small particle size powders with 1 h ball milling process. Nonetheless, more research on the mechanical properties test and thermal conductivity for mould insert is needed. It has been shown that very fine powders have relatively large surface areas and, thus, are highly reactive not just with oxygen, but also with other gases, such as nitrogen [139] or hydrogen [140].

#### 2.1.6. Milling Temperature

The temperature during ball milling is a critical parameter that impacts both the milling process and the quality of the resulting powders. Ball milling is commonly performed at ambient temperature or under cryogenic conditions [75,141,142]. The optimal milling temperature depends on several factors, including the material being milled, the milling duration, the milling environment, and the milling tools. It is generally preferable to mill at temperatures below the melting point or decomposition temperature of the material to minimize the occurrence of milling-induced defects [143,144]. Suarez et al. [42] noted that the duration required to reach the critical dislocation density for dynamic recrystallization is influenced by the ratio of grinding bodies/chips volume in the grinding process of recycled Al-Si-Zn-Mg alloy chips. This is due to the fact that the temperature inside the ball mill jar is dependent on the level of filling of the grinding stock. Collisions during milling can be elastic, leading to particle size reduction and the generation of crystal defects, or inelastic, which raises the temperature of the process and often results in the release of heat energy [119,120,145]. According to Schmidt et al. [146], the maximum temperature (77 °C) is attained when no grinding material is in the jar. Adding a small quantity of quartz sand to be ground significantly reduces the temperature [147]. This is because the added material alters the elasticity of the collisions, thereby slowing the grinding bodies and reducing the amount of energy that is lost as heat. For instance, the temperature increase for milling with three balls with a diameter of 10 mm resulted in a temperature increase of 3 K after 30 min. However, further studies on the implementation of the planetary ball mills procedure are required.

#### 2.1.7. Ball Size Distribution

A key component of the grinding process in a ball mill is the ball size distribution (BSD), which controls the grinding efficiency, product size distribution, and media wear rate. Higher rotational speeds and relatively small ball sizes produce finer BSDs, whereas lower speeds and significantly bigger balls produce coarser ones [71,110,148,149,150]. Moreover, large balls break down coarse feed materials, while smaller balls help in the creation of fine powder by reducing the amount of free space between the balls [151]. The milling ball size is also an important parameter that can impact the final product properties [95,108]. Small balls lead to a higher milling energy and faster particle size reduction, while larger balls lead to a lower milling energy and slower particle size reduction. For example, in a mill with a revolution ratio of 150 mm and a 360 ml jar filled with 3 mm balls, the mean particle size of alumina was reduced from 4.2 μm to 1.1 μm within a 30-min duration at a rotational speed of 400 rpm [152]. The ball milling process was recommended by some researchers for the study, primarily because of the limitation in ball mill feed size (10 mm) and to mitigate the potential occurrence of undesirable reactions that could result from extended ball milling crushing runs [153]. However, larger balls can also lead to improved mixing and homogenization [54,60,90]. Therefore, the material and size of the milling balls are important grinding parameters that can significantly affect particle size. For instance, using balls with a diameter of 20 mm results in coarser particles compared to those with a diameter of 6 mm, as reported in a study [148]. Wang et al. [154] investigated the effect of milling ball size on the microstructure and mechanical properties of a cobalt–chromium–molybdenum alloy (Co-Cr-Mo). They found that using larger balls (10 mm) resulted in better mechanical properties and a more uniform microstructure. In the current study, ball diameters ranging from 5 to 30 mm were investigated, with a 5 mm increment in diameter (5, 10, 15, 20, and 30 mm). These specific ball sizes are commonly employed in experimental setups [155]. Shin et al. [150] conducted an investigation on the influence of rotation velocity (rpm) on the average particle size during a 12 h milling period. The findings revealed that higher rotation velocities led to a finer average particle size, attributed to the increased number of rotations. Specifically, a ball diameter of 5 mm was found to be optimal at 50 rpm, resulting in an average particle size of 2.3 μm. Similarly, a ball diameter of 3 mm was optimal at 100 rpm, yielding a size of 1.4 μm, while a diameter of 2 mm was optimal at 153 rpm, resulting in a size of 0.84 μm. These trends are illustrated in Figure 11. Likewise, smaller balls tend to increase the number of collisions, resulting in a higher total kinetic energy. This, in turn, improves the milling efficiency, leading to a reduction in particle size. The relationship between the number of contact points and ball size is depicted in Figure 12. Based on these results, it can be concluded that the ideal ball size for efficient milling decreases with the rotation speed of the mill.

According to the above review, ball milling is a widely used process and has advantages for grinding materials into fine powders or slurries for various applications [55,124,125]. Critical processing parameters, such as rotation speed, grinding time, and ball-to-powder ratio, can significantly affect the efficiency and outcome of the grinding process [157,158]. A higher ball-to-powder ratio can improve process efficiency but also increases wear on the grinding media and jar [157,159]. However, many previous studies have overlooked the importance of conducting mechanical properties tests and thermal conductivity analyses on the resulting powder samples. Therefore, it is crucial to take into account both the material properties and the desired output when determining the optimal processing settings. Additionally, it is possible to use alternative materials for ball mill jars and balls based on specific application requirements [158,160]. Despite these considerations, the ball mill process remains a versatile and significant tool across various industries. By carefully evaluating and selecting the appropriate media, one can enhance the efficiency and effectiveness of the milling operation.

### 2.2. Selecting Ball Mill Media

The choice of milling media (balls and jars) is a critical factor in the ball milling process, impacting both the milling energy and the final product size. Because of the wear and tear effect of the milling balls on the milling jar walls, it is crucial to use the right materials for the milling media [161]. Several criteria, including shape and size, are interdependent and influence the choice of milling media materials. In ball milling, meeting specific requirements is crucial and must be carefully considered to optimize the milling process. Two primary criteria that the milling media must meet are (i) a large surface area for adequate interface with the milled material and (ii) the necessary weight for providing sufficient energy [162]. This article discusses several crucial factors that must be considered when selecting milling media.

#### Alternative Materials for Ball Mill Jar and Balls

The most challenging feature of a powder particle is its hardness; hence knowing its hardness is crucial when selecting milling media [158,163]. The more efficient the mill, the tougher the milling medium will impact on the quality of final products [164]. Milling with hard materials such as tungsten carbide, zirconium, hardened steel, and agate results in maximum milling performance and the minimum milling time required to obtain small and uniform nanoparticles [82,165,166]. Nonetheless, the use of such a hard milling medium can lead to ball wear, increasing the possibility of foreign material contamination in the milled powders. Materials used to make milling media are listed in Table 1, along with their corresponding Vickers hardness values [161].

Fragmentation in mechanical milling of chips for powder manufacture based on particle size was analyzed. Milling time also affects fragmentation and particle size formation; for example, increasing milling time from 3 to 5 h enhances yield fragmentation [167]. Additives such as PCA; reinforcements such as niobium carbide, vanadium carbide, silicon carbide, or titanium carbide; and milling time and method can all have an impact on the particle size distribution (PSD). However, it was also desirable to develop powders with a nanostructure. Powder composites with a metal matrix are also possible to be made from metal chips. Some writers addressed this issue in the manufacture of duplex stainless steel composite powder by adding vanadium carbide (VC) particles and increasing the milling duration [168]. This study also demonstrated that increasing the reinforcing concentration results in a decrease in average particle size. In contrast, another study demonstrated that the sintered composite has a lower hardness than the as-received alloy and that the porosity of the sintered composite is to blame for the failure [169].

A planetary ball jar would be a container in which materials are ground by revolving a cylinder containing grinding balls. The jar is typically made of ceramic, plastic, or stainless steel used to hold and rotate the grinding media and material being ground [146,170,171]. The size and shape of the jar may vary depending on the application and the amount of material being ground [126,146,147,157,172]. The jars can range in size from a few milliliters to several liters and are often designed to be stackable for easy storage. The grinding balls used in a ball mill are typically made of steel or ceramic materials and come in a variety of sizes and shapes [72,97,146]. The size of the balls can range from a few millimeters to several centimeters in diameter, depending on the application and the desired grinding outcome [120,173]. The ratio of the grinding media to the material being ground, known as the ball-to-powder ratio, can also have an impact on the grinding outcome [72,158,174,175,176]. A larger ball-to-powder ratio typically results in a more efficient milling process but may also lead to increased wear on the grinding media and jar [72,114,146,174,177]. Typically, ball mills are used in the pharmaceutical, chemical, and metallurgical industries to grind materials into fine powders or slurries prior to further processing or analysis [150,177,178].

There are several alternative materials that can be used for ball mill jars and balls, depending on the application and desired properties. Some common alternatives are:i.Agate: Agate is a naturally occurring stone that is often used as a substitute for ceramic or porcelain jars. It is known for its hardness and durability, making it ideal for grinding hard materials [151,179,180].ii.Tungsten Carbide: Tungsten carbide is a hard and dense material that is often used as a substitute for steel balls. It is highly resistant to wear and abrasion, making it ideal for grinding materials that are difficult to grind [181].iii.Stainless Steel: Stainless steel is another popular material for ball mill jars and balls due to its resistance to corrosion and chemical reactions. It is often used in applications where the material being ground is reactive or corrosive [154,171,181,182].iv.Alumina: Alumina is a ceramic material that is often used as a substitute for porcelain or glass jars. It is highly resistant to wear and corrosion and is often used for grinding materials that are abrasive or highly acidic [96].v.Zirconia: Zirconia is a ceramic material that is highly resistant to wear and abrasion. It is often used as a substitute for steel balls in applications where the material being ground is highly abrasive or reactive [96].

Based on the findings of previous researchers [67,94,106,147,156,160,162,182] concluded that the selection of ball mill media has a significant impact on the ball milling process, particularly when refining small particles. The choice of media is influenced by the type of raw material being processed. However, further research can be conducted to explore the ball milling of recycled copper chips for mold insert of RT application. The effects of various ball mill parameters on mechanical properties are discussed in more detail in the following section.

### 2.3. Influence of Ball Mill Parameters on Mechanical Properties

In the field of materials science and engineering, ball milling is a common mechanical technique used to synthesis or otherwise alter a wide variety of starting materials. In order to optimize milling processes and achieve desired material characteristics, it is essential to understand the influence of ball mill settings on the mechanical properties of milled materials. This literature review explores the effects of changes in ball mill jar size, ball-to-powder ratio, and process control agents (PCA) on mechanical attributes.

#### 2.3.1. Impact of Different Ball Mill Jars on Mechanical Properties

Ball milling is a widely used technique for the preparation and modification of various materials. The capabilities of the ball mill jars used throughout the milling process significantly impact the mechanical characteristics of the finished goods. The type of jar material, shape, and size can all significantly impact the milling efficiency and resulting mechanical properties of the milled products. Table 2 summarizes previous studies employing different ball mill jars [69,77,78,79,103].

Enayati et al. [69] presented their study in 2007 on the production of nanocrystalline stainless steel powder using ball milling. Thus, the as-received chips were small, discontinuous, and non-fibrous, with an average 2–4 mm length and a C-shaped morphology. Before ball milling, the authors observed that the resulting chips were washed with acetone to remove the oil. The particle morphology of the resulting powder was influenced by the size of the stainless steel swarf, the milling period, and the ball-to-powder ratio. The study presented exciting findings, and the methodology and data analysis used by the authors appeared to be sound. After 50 and 100 h of ball milling, the mean particle size was 300 μm and 60 μm, respectively. The powder particles achieved a microhardness value of 820 Hv after 50 h of ball milling, which subsequently rose to 850 Hv after 100 h of ball milling. However, after undergoing annealing, the microhardness of the ball-milled powder decreased to 630 Hv and 710 Hv, respectively. These findings suggest that the ball milling approach can efficiently produce nanocrystalline stainless steel powders from scrap chips without the need for an additional annealing process. However, one limitation is that the authors did not comprehensively of mechanical properties testing, such as its compressive strength and thermal conductivity measurement. This information is important for understanding the potential applications of the produced powder. Another limitation of the article is that the authors did not provide a comparison of their results with other studies on the production of nanocrystalline recycled copper chips using ball milling. This would help to evaluate the novelty and significance of their findings. Overall, while the study presented interesting findings, a more comprehensive characterization of the produced powder and a comparison with other analyses would have strengthened the article.

Furthermore, Liang et al. [77] studied the ball milling method, refining method, and mechanisms of tungsten powder. The authors conducted a series of experiments using a planetary ball mill to refine tungsten powder and investigated the impact of different grinding settings, such as grinding duration, ball-to-powder ratio, and rotational speed, on the refinement process. The experimental results showed that planetary ball milling significantly refined tungsten powder and decreased its average particle size. The authors found that longer grinding, higher ball-to-powder ratio, and fast rotating speed lead to a more significant refinement effect. They also investigated the mechanisms of the refinement process and found that the primary mechanism was plastic deformation, and the secondary mechanism was fracturing and cold welding. The powder extracted from the vials after 50 h of ball milling exhibits nanoscale particles ranging in size from 0.1 µm to 0.12 µm. Figure 13 depicts the correlation between the average particle size of W and the ball milling period, as reported. It demonstrates that initial particles are welded to a sheet by impact force and that the sheet becomes thinner and bigger due to the squeezing action. The sheet is shattered into little fragments as the milling time continues to grow, as stress concentration causes fractures to form, and the sheet is then split into small pieces. After a longer period of ball milling, the fragments are reduced to considerably smaller bits, and then they enter the dynamic balance stage, where the particle size is stabilized. Overall, the article provides valuable insights into the process and mechanisms for refining tungsten powders through ball milling. The experimental results are well-presented and supported by appropriate data and analysis. Nevertheless, the study has some limitations that should be considered. For example, the study was limited to the rotation speed of the ball mill, and the size of the balls was not mentioned in this article. Moreover, the study did not examine the impact of different milling durations on the effectiveness of the resulting mechanical properties and thermal conductivity measurement. Additionally, the authors did not provide a detailed discussion of the potential practical applications of the refined tungsten powder.

An exploration of the mechanical milling synthesis of B_4_C particle-reinforced Al2024 composites were reported by Gallardo et al. [78]. The researcher investigated the influence of milling duration with B_4_C content on the resulting composite powder properties and the sintered bulk composites’ microstructure and mechanical properties. The results revealed that the B_4_C particles were uniformly dispersed in the Al2024 matrix, and the particle size reduced with the milling time increases. Adding B_4_C particles improves the composite microhardness in Figure 14a, while the maximum strength in Figure 14b and ductility are slightly reduced. The optimum milling time and B_4_C content to balance strength and ductility are reported. Overall, the research provides useful insights into the fabrication and properties of B_4_C particle-reinforced Al2024 composites using mechanical milling. The average particle size of Al2024-2 wt.% B_4_C was found in 20 µm after 2 h of milling. However, the particle size of Al2024 mixed with other concentrations of B_4_C was not clear in this study. Moreover, the study only investigates the properties of the sintered bulk composites but does not provide a detailed analysis of the thermal conductivity and the resulting composite powders. Another limitation is the absence of a comparison with other methods for fabricating B_4_C particle-reinforced Al2024 composites, which could help to evaluate the advantages and disadvantages of mechanical milling.

Subsequently, Sarah et al. [103] explored the possibility of recycling aluminum scarps into gritty powder using ball milling for metal injection molding (MIM) applications. The authors conducted experiments to determine the optimal milling duration (7 h) and rotation speed (370 rpm) for the milling and analyzed the resulting powder for its properties and suitability for MIM. The resulting powder had a smaller particle size and improved flowability compared to the raw chips and was found to have good sinterability and mechanical properties for MIM. Overall, the study provides promising results for using ball milling in recycling aluminum chips into fine powder for MIM applications. However, there are some limitations associated with the study. For example, the experiments were conducted on a small scale, and whether the same results can be achieved on a larger scale is unclear. Additionally, the study focused only on the ball milling of the resulting powder and did not investigate the mechanical properties of hardness, compressive strength, and thermal conductivity performance. Nonetheless, the study provides valuable insights into the potential of using ball milling for recycling copper chips and improving the sustainability of metal manufacturing processes.

Li et al. [79] investigated the effect of different environments on the morphology of nickel powder produced by the ball milling process. This experiment with two different types of ball mills and studied the impact of grinding duration, grinding rotation velocity, and milling conditions on the morphology of the resulting nickel powder. The powder size and particle distribution obtained were characterized using the Mastersizer 3000 (Malvern Panalytical Ltd. from Enigma Business Park, Grovewood Road, Malvern WR14 1XZ, United Kingdom.). The results, as depicted in Figure 15, demonstrate a mean particle size of 28.4 ± 0.5 µm. Furthermore, the red line on the graph denotes the particle size range between 1 µm and 100 µm, as specifically indicated in this study. The study is well-organized, and the experimental design appropriately addresses the research question. In addition, when ethanol was utilized as the milling medium, tiny, evenly spherical particles of nickel powder were produced. The results are presented clearly and in detail, and the conclusions drawn from the study were supported by relevant data. Figure 16 shows the particle size change in the ambient air as a function of speed. When the milling velocity increases from 100 rpm to 500 rpm, the particle size of the Ni powder reduces from 57.5 nm to 12.2 nm, and the lattice micro-strain improves from 0.001% to 0.131%. The findings are novel and can benefit researchers and engineers working on powder metallurgy, especially in nickel powder production. The study only looks at how the environment affects the shape of nickel powder. Other things that can change the characteristics of the powder, such as the kind and amount of process control agents and the properties of the starting material, were covered in their scope of research. Unfortunately, the experiments did not elaborate on the mechanical properties and thermal conductivity that underlie the observed effects, so more studies are needed. This study can be utilized as a starting point for further investigation into the interaction between the milling environment and nickel powder morphology. However, further analysis is essential for fully comprehending how other factors affect powder characteristics when utilizing ball milling jars made of different materials.

Based on the aforementioned review, the manufacturing of new potential in producing small particle powder materials relies on several critical aspects, including microhardness and compressive strength, which vary depending on the ball mill jars (high chromium steel, zirconium, and tungsten carbide) used. Furthermore, it was observed that the hardness of the samples significantly increased after the ball milling process [58,164]. However, most previous studies have overlooked the evaluation of thermal conductivity in specimens after ball milling, particularly when attempting to determine the optimal material proportions for manufacturing mold inserts. Understanding the effects of different ball mill jars on mechanical properties can help researchers optimize the milling process and tailor the characteristics of the final product for specific applications.

#### 2.3.2. Effect of Different Ratios of Ball Mill on Mechanical Properties

The current study investigates the type of ratio for ball mill jars listed in Table 3 and [63,80,168,181,183]. Different ball-to-jar ratios can affect the milling kinetics, the size, and distribution of the milled particles, and ultimately the mechanical properties of the resulting materials [103,105,114,141,156,163,184,185,186]. Thus, it is essential to comprehend how various ball-to-jar ratios affect the mechanical characteristics of the milled materials to optimize the ball-milling procedure and create novel materials with specialist qualities.

In 2014, Biyik and Aydin [63] examined the influence of the quantity of process control agent (PCA) on the properties of Cu25W composite powder produced through the mechanical alloying process. The study discusses the impact of PCA on the particle size distribution, morphology, and microstructure of the Cu25W composite powder. The milling container utilized in this study was equipped with grinding balls measuring Ø10 mm, which were manufactured using tungsten carbide (WC). The results showed that the quantity of PCA has a significant influence on the powder’s final characteristics. One potential limitation of the study is that it only examined the impact of PCA on one specific type of composite powder. Further research may be needed to determine whether the findings can be extended to other types of composite powders. Furthermore, after 18 h of milling, there was a slight decrease in the average particle size, with the size dropping below 20 µm for each powder batch. However, it is important to note that the study does not offer a detailed explanation of the mechanism by which PCA (Polymeric Carboxylic Acid) influences the properties of the powder. This information could be useful for future research and development in the field. Overall, the study is a valuable contribution to the field of e-waste recycling and the use of ball milling for metal recovery. However, additional research into selecting the various rotation speed of the ball mill is needed. Moreover, the difference in BPR has to be proposed in terms of the quality of the small particles for final products.

Next, Guaglianoni et al. [80] presented an investigation into the production and characterization of WC-12 wt% Co nanocomposites through a high-energy ball mill process. The researchers evaluated the impact of milling duration and ball-to-powder ratio on the nanocomposites’ characteristics. The milling duration had a significant effect on the particle morphology, and ball to powder ratio of 1:20 was identified to be acceptable. An advantage of the study is that the synthesized nanocomposites were thoroughly characterized using techniques such as scanning electron microscopy (SEM), X-ray diffraction measurement (XRD), and energy dispersive spectroscopy analysis (EDS). The results obtained through these techniques provided a comprehensive understanding of the properties and structure of the nanocomposites. The average particle size obtained was 1.63 µm with 500 rpm rotation velocity and 5 h of ball milling time. However, the study has a few limitations. Other parameters, such as the type of milling media used, may have had a significant effect on the properties of the nanocomposites, but the authors did not explore these. The evaluation of mechanical properties, including compressive strength, density, hardness, as well as thermal conductivity, is crucial to assess the applicability of nanocomposites for practical purposes. However, there is currently a lack of comprehensive data regarding the performance of these properties, highlighting the need for further investigation. Overall, the study provided useful information regarding how a high-energy ball mill can be used to make [98] and describe WC-12 wt% Co nanocomposites. However, further study is required to completely comprehend the impact of varied milling settings and to determine the mechanical characteristics of the resulting nanocomposites.

In 2018, Petrovic et al. presented [183] an optimization study of the nanoparticle ball milling process parameters using the response surface method (RSM). The study aimed to optimize the process parameters to improve the size of reduction and dispersion of the nanoparticles. The authors applied a factorial design to determine the impact of the grinding duration, grinding rotation speed, and ball-to-powder ratio on nanoparticle size and degree of agglomeration. Table 4 presents the minimum and maximum values of the variables, as well as the complete experimental design in both real and coded forms, with respect to these values.

The results indicate that the grinding time and ball-to-powder ratio exert the greatest influence on nanoparticle size and degree of agglomeration, whereas milling speed has a relatively minor effect. The authors obtained an optimized set of process parameters that produced nanoparticles with a small average size and good dispersion. Overall, the study provides useful information for optimizing the ball milling process for nanoparticle production. However, the study has some limitations, including the fact that it was conducted on a specific material system and that the results may not be directly applicable to other materials. Additionally, the study did not consider the effect of other parameters that could impact the process, such as the type and size of the grinding media, the type and size of the starting material, and the presence of process control agents. In order to completely comprehend the implications of these factors and optimize the ball milling process for various material systems, more studies are required. However, the study has a few limitations. Firstly, the milling environment, which can have a major effect on powder quality, was not investigated. Secondly, the study did not examine the powder’s characteristics, such as the influence of different ball-to-powder ratios on mechanical properties and thermal conductivity. Lastly, it should be noted that the study was focused solely on metal powder, and further research is needed to assess the applicability of the findings to the refinement of other types of metal chips.

In the same year, Vasamsetti et al. [181] examined the manufacture of nanoparticles utilizing a planetary ball mill machine and the adjustment of milling settings for enhanced particle size reduction. The authors determined the best combination of milling parameters, including rotation speed, ball size, milling duration, and charge ratio, using a response surface methodology (RSM). In this study, milling was performed with a vertical planetary ball mill that had a tungsten carbide (WC) milling bowl jar and WC balls. Through a successful synthesis of nanoparticles using the planetary ball mill, the study revealed that optimized milling parameters led to smaller particle sizes and increased yield. The authors also employed various analytical techniques to confirm the formation of nanoparticles, including X-ray diffraction measurement (XRD) and transmission electron microscopy (TEM) analysis. The optimum ball mill parameters to achieve a small particle size of 0.056 µm include a ball mill speed of 500 rpm, a milling time of 10 h, and a ball-to-powder ratio (BPR) of 10:1. These settings are sufficient for obtaining better results. However, the study does have certain limitations that could be addressed in future research. Firstly, the investigation only focused on non-metal samples for the ball mill experiments. Additionally, the absence of mechanical strength tests and the inability to measure thermal conductivity due to the non-conductive nature of the material are notable shortcomings. Therefore, this study offers valuable insights into optimizing milling parameters for nanoparticle synthesis using a planetary ball mill, and further studies are needed to explore the impact of other variables on the particle size reduction and to generalize the findings to different types of metal chips for the ball mill process.

In another report, Mendonça et al. [168] examined the effect of ball mill setting on the recycling of stainless steel chips using ball milling. The results of the study showed that milling duration and ball-to-powder ratio had a significant effect on powder production, while milling speed had a lesser effect. The authors also observed that prolonged milling duration and high ball-to-powder ratios led to a minimization in the particle size of the stainless steel powder but also resulted in the introduction of impurities in the powder. As seen in Figure 17, a more aggressive milling condition, characterized by a rotational speed of 350 rpm, a BPR of 1:20, a milling duration of 50 h, and the addition of 3% VC, resulted in a significant reduction in particle size. The average particle size ranged from 25 to 135 μm. Overall, the article presents a systematic approach to investigate the impact of different ball mill setting on the recycling of stainless steel chips using ball mills. Using a full factorial design approach provides a comprehensive understanding of the impact of milling setting on powder production. The findings may be utilized to optimize milling settings for the generation of stainless steel powder from machining chips. However, the article lacks detailed information regarding the type of ball mill jar and the size of the ball mill employed in the study. These parameters play a crucial role in influencing the performance of the ball mill during the refining process of the samples. Additionally, important tests such as compressive strength and thermal conductivity were not conducted for the recycled stainless steel powder. Addressing these limitations in future research would provide a more comprehensive understanding of the ball milling process and its impact on the properties of recycled metal chips.

Five (5) studies related to the use of ball milling for composite and nanoparticle synthesis were summarized. Biyik and Aydin [63] examined the impact of the amount of process control agent (PCA) on the properties of Cu25W composite powder, while Guaglianoni et al. [80] investigated the production and characterization of WC-12 wt% Co nanocomposites through high-energy ball milling. Petrovic et al. [183] presented an optimization study of the nanoparticle ball milling machine settings using the response surface method (RSM), while Vasamsetti et al. [181] examined the manufacture of nanoparticles using a planetary ball mill machine and the adjustment of milling settings for enhanced particle size reduction. Mendonça et al. [168] investigated the influence of milling duration, ball-to-powder ratio (20:1), and milling rotational speed on the production of recycled stainless steel powder from machining chips. They employed a complete factorial design technique to optimize the ball mill conditions and found that these specific parameters led to a small particle size of 135 µm and below. This study provides valuable insights into the effects of different milling parameters on composite and nanoparticle synthesis. However, further research is necessary to fully assess the compressive strength, thermal conductivity, and overall effectiveness and economic viability of these methods for industrial applications, particularly in the field of injection molding. Additional investigations will contribute to a better understanding of the properties and potential utilization of the synthesized materials in the context of industrial-scale injection molding.

#### 2.3.3. Effect of Different Process Control Agents (PCA) of Ball Mill Process

Ball milling is a commonly used technique for the preparation of various materials, including chemicals, pharmaceuticals, and minerals. It has also been documented as a pretreatment approach to advance the development of composites using alternative techniques or to subject the obtained copper powders to subsequent milling under different conditions than the initial process [166,187,188]. The use of process control agents (PCA) significantly impacts the materials produced through ball milling [189,190]. This paper aims to examine the effects of various process control agents used in ball milling. The role of PCA is to prevent undesirable consequences such as particle adhesion to the jar and media, particle agglomeration and coarsening, and excessive cold welding, ensuring optimal process outcomes as summarized in Table 5 [47,105,148,191,192].

Initially, Zhang et al. [191] presented a study on the use of ball milling to recover valuable metals from electronic waste (e-waste). The authors explored a method to extract valuable metals from e-waste swarf, which can be a source of pollution and waste. The study results showed that the ball milling process was effective in breaking down the e-waste swarf and liberating valuable metals, such as silver, copper, and gold. The authors also found that the ball milling process is environmentally friendly compared to traditional methods, as it reduces the use of chemicals and produces less waste. The article provided a clear explanation of the methodology. Figure 18 depicts the morphological changes in the specimen before and after the ball milling procedure. The authors presented a complete analysis of the e-waste swarf and the recovered metals, which is vital for evaluating the success of the ball milling process. This study on ball mill conditions avoids the use of nitric acid or mixtures containing nitric acid, sulfuric acid, or hydrochloric acid. Therefore, it effectively prevents the release of gaseous pollutants. Moreover, the researcher indicated the potential of ball milling as an environmentally benign approach for extracting precious metals from e-waste. Despite its significant findings, the study has some limitations that require attention. Firstly, the authors do not offer a detailed economic analysis of the process, which is crucial in determining the method’s feasibility for large-scale industrial applications. Secondly, the effect of ball milling on the compressive strength, hardness, and thermal conductivity of the waste powders was not discussed in detail in this study. Addressing these limitations could improve the overall understanding of the ball milling method and its potential as a viable solution for e-waste management.

Fullenwider et al. [148] described a technique for recycling and machining scrap chips to make metal-additive manufacturing-compatible powder. Two stages of the ball milling process were proposed, which involve a first stage of wet ball milling and a second stage of dry ball milling to produce a fine powder. The results showed that the ball milling process effectively produces a high-quality powder suitable for use in powder bed fusion additive manufacturing. Figure 19 illustrates the relationship between ball diameter and final particle morphology when the maximum strain depth is normalized by the particle diameter at the optimal particle size range for LENS (38–150 μm). The homogenized maximum deformation depth of a particle with a 100 μm diameter occurred because a 20 mm ball is 81 μm of the particle’s diameter. In comparison, the diameters of the particles caused by the effects of a 10 mm ball and a 6 mm ball are 28 μm and 13 μm, respectively. When compared to machining chips, the hardness of powder particles that have been ball-milled rose by 44%. This is due to the austenite to martensite phase transformation as well as additional strengthening processes, such as grain size refinement and higher dislocation density. The powder generated by the procedure was reported to have a limited size range and a homogeneous morphology, both of which are crucial for generating superior powder flowability and packing density. However, a further investigation of the compressive strength and thermal analysis is required.

Next, the development of a low-cost copper (Cu)-based composite material enhanced with silicate glass particles for thermal applications was reported by Prosviryakov et al. [47]. The composite was produced using mechanical alloying, a technique in which particles are combined and then exposed to high-energy ball milling. The authors examined the impact of varied milling times on the composite’s microstructure, hardness, and thermal conductivity. According to the results of the investigation, the inclusion of silicate glass particles greatly enhanced the composite’s hardness and thermal conductivity. The authors found that longer milling times led to smaller particle sizes, more homogeneous mixing, and higher thermal conductivity. With increased milling time, the microhardness of the composite powder particles increases to around 320 HV, but dislocation density and copper grain size decrease. Moreover, the researchers developed light microscopy images of Cu-glass composite between 1 and 7 h (particles size below 100 µm) of the ball mill. Finding from this study, the optimized composite had a thermal conductivity of 200 W/mK, which is much higher than that of pure copper. However, further studies on the corrosion resistance and tribological test of samples are required. 

Bhouri and Mzali [192] studied how ball mills and hot compaction impact the microstructure and physical characteristics of recycled aluminum alloy (2017) powders. The powdered recycled aluminum alloy powder was milled for different time periods before hot compaction at varying temperatures and pressures. The results showed that the microstructure of the recycled aluminum alloy powder was refined with increasing milling time, and densification improved with increasing hot compaction temperature and pressure. The sintered density of recycled powder rose as compaction pressure and time holding were increased after hot compaction under 150 MPa. The sample milled for 20 h manifests irregular microstructure and large pore distribution due to insufficient external pressure. Nevertheless, the enhancement in pressure levels was sufficient to produce a homogeneous shape during the subsequent ball mill, which led to less coarsening of the structure. Due to insufficient plastic deformation when applying low pressure, the effective contact area of 2017 aluminum alloy powder was reduced [193]. The authors also observed an increase in the hardness and electrical conductivity of the compacted samples with increasing hot compaction temperature. The study presented interesting results that contribute to understanding how ball mills and hot compaction processes impact the properties of recycled aluminum alloys. The article discussed various milling times for the ball mill strategy. However, this particular study focused specifically on one rotation speed (250 rpm). In contrast, previous researchers, such as Zhang et al. [191], conducted experiments covering a broad range of speeds, ranging from 300 to 600 rpm. When the ball mill operates at higher rotation speeds, it can facilitate the effective chipping or fragmentation of the material being processed. Moreover, the article only focused on the electrical and mechanical characteristics of the recycled aluminum alloy powder and neglected other important aspects such as corrosion resistance, thermal properties, and Orowan looping mechanism. A more comprehensive analysis would provide a better understanding of the potential applications and limitations of using recycled aluminum alloys in various engineering fields. In summary, the study provided valuable insights into the impact of ball mills and hot compaction on the microstructural and physical characteristics of recycled aluminum alloys. However, the inclusion of additional information on the experimental methods and the use of another recycled metal material, such as copper chips, for comparison purposes would greatly enhance the significance of the research, particularly in relation to its application in mold insert scenarios.

In addition, Dias et al. [105] examined the impact of ball milling settings on the microstructure and magnetic characteristics of vanadium carbide-reinforced aluminum bronze swarf. The effect of milling duration, milling velocity, and medium grinding size on the composites’ microstructure and magnetic properties was explored using a factorial design in this work. Incorporating VC is the most significant component that augments the energy level infused into the system in the DOE analysis. This is because of its greater hardness, which initiates a ductile–brittle contact mechanism among particles. Because of collisions between the milling balls and the vial walls, the soft metal particles tend to amalgamate and form bigger particles in the early phases. A milling speed of 350 rpm, 3% VC, and a milling time of 50 hrs all helped to speed up the size reduction of the particles, as shown in DOE tests 8 values of 6.57 μm. The results indicate that increasing the ball milling duration and milling speed leads to a decrease in the particle size of vanadium carbide and an increase in the number of smaller carbide particles [194]. The researchers discovered that using larger grinding balls led to an increase in carbide particle size. Additionally, the magnetic characteristics of the composites made by ball milling for longer periods were better. Using a factorial design is an effective method for determining how various factors influence the characteristics of composites. PCA played a crucial role in the process as a deagglomerator during milling, effectively preventing cold welding between powder particles. It functions by being absorbed onto the particle surfaces, thus inhibiting their agglomeration. However, it is worth noting that the study did not specifically investigate the compressive strength and thermal conductivity of the ball-milled chips at each stage. Additionally, the effects of ball milling on recycled copper chips were not investigated. Nevertheless, this article lays a strong groundwork for future research on the impact of ball mill settings on the microstructure and properties of metal matrix composites. Further studies are necessary to comprehensively comprehend the relationships between various variables and the characteristics of the composites obtained.

In conclusion, Zhang et al. [191] investigated the feasibility of using ball milling to recover precious metals from discarded electronics. Researchers discovered that the process is both efficient and kind to the environment. By using a two-stage ball milling process, Fullenwider et al. [148] detailed a method for recycling machining scrap chips into a metal additive manufacturing-compatible powder. Ball milling was used by Prosviryakov et al. [47] to create a low-cost composite material based on copper and reinforced with silicate glass particles for thermal purposes. Furthermore, Bhouri and Mzali [192] explored the effects of milling and hot compaction on recycled aluminum alloy powders’ microstructure and physical properties. Moreover, Dias [105] indicated that increasing the ball milling duration and milling speed leads to a drop in the particle size of vanadium carbide and an increase in the number of smaller carbide particles.

Records exist showing how the size of powder varies when various materials are mechanically processed [58,62,186,189,195,196,197,198]. According to previous research [49,187,190,197,199,200], it has been established that increasing the milling time leads to a reduction in the mean grain size. Furthermore, this review study reveals that the rotation speed of the ball mill should not surpass 600 rpm to prevent excessive heating and potential thermally induced reactions between the specimens and the milling media (jar and ball). Although some papers lack essential details such as an economic analysis or information on experimental procedures, they all provided interesting insights and findings that contribute to a better understanding of ball milling processes, especially for microstructure and properties measurement.

### 2.4. Evaluating the Properties of Metal-Chipped Powder Particles

Chips from metallurgical processes, including turning [62,201], roughing [168,202], milling [99], and finishing [202], can be utilized in recycling [103], or they can be freshly machined [203] for a new purpose. Aluminum alloys [103], stainless steel [69], tool steel [204], titanium alloy [204], tin alloy [71], and nickel alloy [205] have all been investigated thus far, albeit in a limited capacity and with varying starting dimensions. Evaluating the properties of these particles is essential to ensure the qualities and performances of the final product. Other analyses, such as structural, surface roughness, electromagnetic, Orowan looping mechanism [198], microwave properties [206], tribological [99], piezocatalytic measurement [207], thermal conductivity [208], corrosion process [209] and hardening, should be considered to evaluate and investigate the milled samples. Moreover, understanding mechanical qualities such as hardness, tensile strength, and toughness is vital in establishing the viability of the material for various applications [45,93,94,97,122,145,210,211]. Nazri et al. [212] research showed that industrial mill-scale waste has the capacity to generate a large number of magnetite (Fe_3_O_4_) particles. In this study, Maity et al. [213] synthesized semiconducting ZnAl_2_O_4_ compound is well-suited for use in optoelectronic devices because to its high optical quality, high electrical conductivity, and low dielectric constant value. Based on the Rietveld analysis and high-resolution transmission electron microscopy (HRTEM) image, it was determined that the average size of the spherical crystallites in the powder milled for 4 h is approximately 17 nm. Liu et al. [214] successfully prepared a high-entropy alloy (HEA) called FeCoCrNiMnTi_0.1_C_0.1_ (TiC10 alloy) using the mechanical alloying (MA) technique and vacuum hot-pressing sintering (VHPS) process. The powder was ball milled in a stainless-steel vial with stainless-steel balls (diameters Ø5, Ø10, and Ø15) under high-purity argon at 250 rpm for 45 h at 250 rpm. When the TiC10 alloy was sintered at 900 °C, it exhibited favorable comprehensive mechanical properties. The alloy demonstrated a yield strength of 1652 MPa and a hardness of 461 HV. These results indicate the excellent mechanical performance of the TiC10 alloy at this specific sintering temperature. However, this investigation did not discuss of thermal conductivity on the final product.

Additionally, assessing the qualities of metal-chipped powder particles is critical for identifying the attributes of the materials and their appropriateness for diverse applications. Furthermore, many researchers studied particle morphology evolution for metal powders and composites using the mechanical milling process [59,188,199,200,215]. Huang et al. [215] investigated the effects of ball milling on the development of the morphology, properties, and microstructure of pure copper powder and other ductile metals. Ball milling is a complex process that can significantly alter the morphology, structure, and properties of materials. Therefore, various analytical techniques such as X-ray diffraction [40,80,130,211,216], scanning electron microscopy [97,159,178,181,217], energy-dispersive X-ray spectroscopy [84,105,121,125], and transmission electron microscopy [47,207,214,216,218] can be used to study the physical, chemical, and structural properties of the particles. However, the findings of previous researchers do not include a comparative analysis of the ball milling process with other existing methods of recycling copper metal waste. Such a comparison would help in studying and understanding the relative advantages and disadvantages of ball milling in comparison to alternative approaches. Without this analysis, it is challenging to evaluate the significance of the proposed process in the broader context of metal waste recycling. Therefore, reviewing the properties of metal chip powder particles is essential for the development of materials science and engineering [37,170] and can result in the creation of novel and creative materials for a range of applications.

## 3. Summary and Future Works

In summary, it is crucial to acknowledge that the raw materials employed in those investigations varied, encompassing composite powder, non-metal powder, and metal scrap derived from ball milling experiments. Furthermore, the absence of original experimental data or results could be attributed to confidentiality and privacy concerns within an industrial environment. By addressing these limitations and conducting further research, scientists can enhance the reliability and validity of their conclusions regarding the effectiveness of ball milling procedures. Therefore, a novel method of recycling copper chips and transforming them into tiny particles using a ball mill needs further investigation. In future trends, researchers are currently exploring ways to improve this method and utilize it in the synthesis of innovative materials with unique characteristics and functionalities [74,129,158]. Additionally, including industrial case studies or collaborating with industry partners (with appropriate confidentiality agreements) can help bridge the gap between academic research and real-world applications.

Ball milling can purpose produce large volumes of metal powder from metal waste. The ball milling process was scaled up to have large quantities of metal powders, depending on the size and capacity of the ball mill. In the case of metal waste, the volume of metal powder produced depends on the quantity and size of the waste material being processed and the efficiency of the ball milling process. However, it is essential to note that the size and capacity of the ball mill used for processing the metal waste will significantly impact the quantity of metal powder that can be produced. Large-scale ball milling processes can be energy-intensive and time-consuming, and the process may need to be optimized for each specific type of metal waste being processed. Additionally, the cost of the equipment and the required maintenance can be a factor in the economic feasibility of using ball milling to produce large volumes of metal powder from metal waste. Overall, ball milling can be a useful technique for producing metal powders from metal waste, and the process can be scaled up to produce large volumes of metal powder. However, the process must be optimized and scaled appropriately to ensure efficient and cost-effective production. Based on the gaps found in the literature, recommendations for future studies are as follows:i.Investigate the effects of ball jar and ball mill size and capacity on the quantity of metal powder produced. This will help identify the optimal ball mill size and capacity for processing different types of metal waste.ii.Study the effects of ball milling settings, such as milling duration, rotation speed, and ball-to-powder ratio, on the quality and quantity of metal powder produced. This will help optimize the ball milling process for each specific metal waste type.iii.Explore additives during ball milling to improve the quality and quantity of metal powder produced. Additives such as surfactants, dispersants, and lubricants may improve the efficiency of the ball milling process and increase the yield of metal powder.iv.The performance of recycled copper powder after ball milling in terms of thermal conductivity, compressive strength, density, thermal diffusivity, and surface roughness must be studied before they can be used as mold inserts for RT application.v.Investigate the environmental impact of ball milling on metal powder production from recycled copper chips (waste). This includes studying the energy consumption, carbon footprint, and waste generated during the ball milling.

In conclusion, this review has provided a clear reference for the future development of the ball milling technique advantages to produce metal powder from machining chips. Thus, an initiative needs to be taken to conduct an analysis of the effect of incorporating metal particles after ball milling as mold insert material for RT and its relationship with compressive strength and thermal conductivity. Moreover, the development and commercialization of low-cost materials for rapid tooling (RT) mold inserts using recycled metal from machining are promising. This approach has shown potential in enhancing the molding efficiency and manufacturing of plastic components, with the aim of developing novel products. These changes ultimately result in desired material properties, including improved mechanical strength and enhanced thermal conductivity. Additionally, exploring the potential applications of these metal powders, including copper, with diverse particle sizes [106] in industries such as additive manufacturing, electronics, and energy storage would be beneficial. Future research should focus on developing cost-effective and sustainable ball milling methods using renewable energy sources to reduce carbon footprint. Greener approaches and evaluating recycling processes for copper particles in mold inserts can improve recycling efficiency and expand ball milling applications in metal processing. Ultimately, this research could lead to the discovery of new, sustainable, and green materials that could greatly benefit the molding and rapid prototyping industries while providing environmentally friendly alternatives.

## Figures and Tables

**Figure 1 materials-16-04635-f001:**
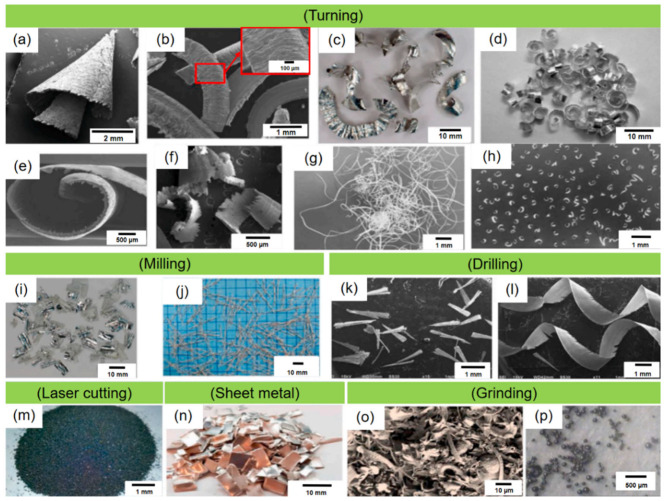
Various shapes of metal chips produced from machining processes [9]: (**a**) grey cast iron [13]; (**b**) stainless steel [14]; (**c**) AA6060 aluminum alloy [15]; (**d**) AZ31B magnesium alloy [16]; (**e**) Ti–6Al–4V [17]; (**f**) CuSn10 bronze [18]; (**g**) chips produced without chip breaker [19]; (**h**) chips of aluminum created by the turning tool’s chips breaker [19]; (**i**) AA6060 aluminum alloy [15]; (**j**) AC4CH aluminum alloy [20]; (**k**) modulation-assisted drilling of stainless steel [21]; (**l**) conventional drilling of stainless steel [21]; (**m**) low carbon steel [22]; (**n**) AA1050 aluminum alloy [23]; (**o**) AISI 4340 steel [24]; (**p**) low alloy steel [25].

**Figure 2 materials-16-04635-f002:**
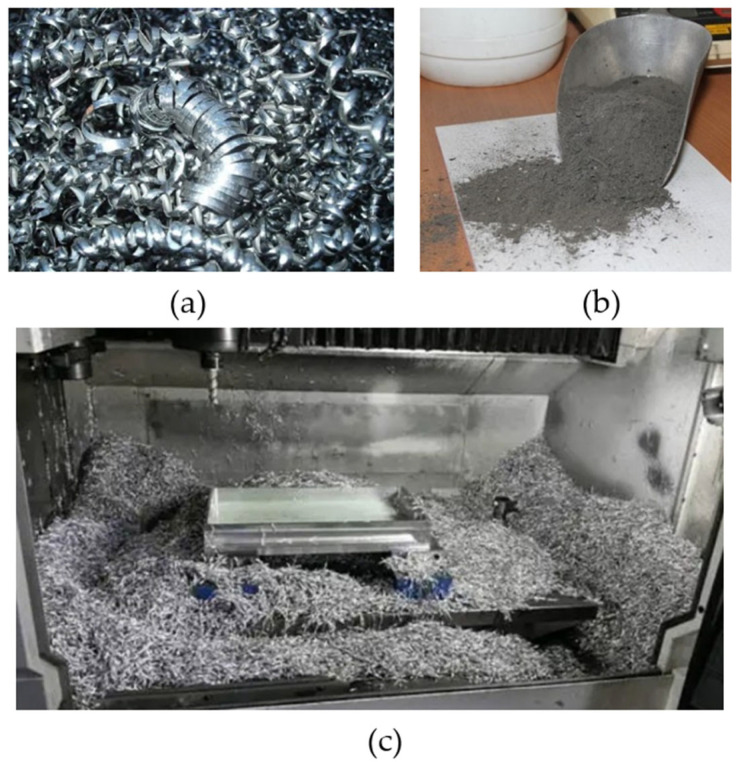
Metal scrap: (**a**) scrap metal from turning; (**b**) scrap metal from grinding; (**c**) scrap metal from milling [48].

**Figure 3 materials-16-04635-f003:**
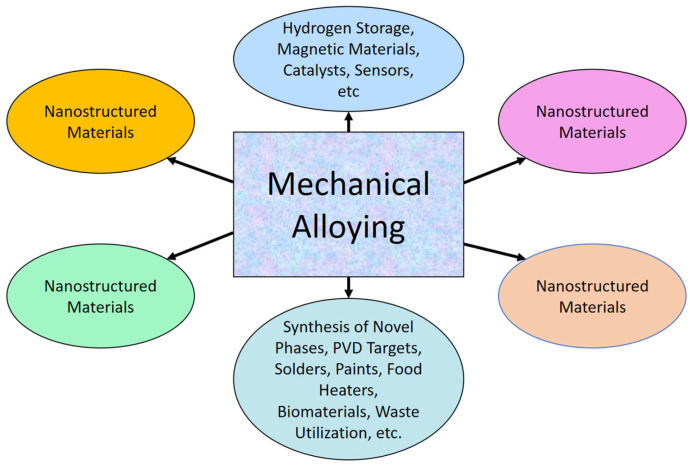
An overview of the applications of the MA process [53].

**Figure 4 materials-16-04635-f004:**
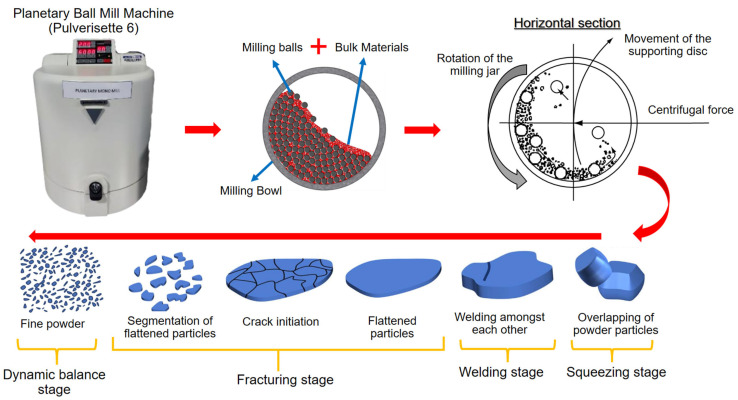
Principles of the ball milling method for reducing powder particle size [9].

**Figure 5 materials-16-04635-f005:**
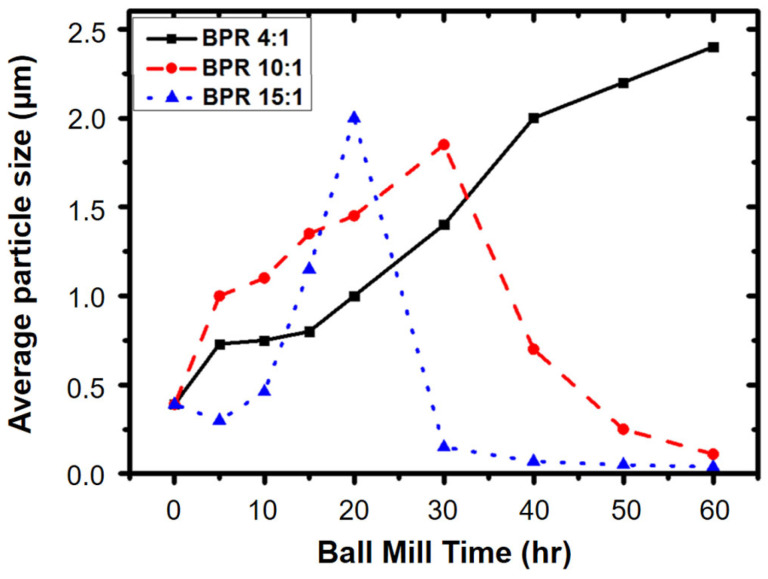
The tungsten powder refinement process corresponding to the changes in tungsten particle size at different milling times [117].

**Figure 6 materials-16-04635-f006:**
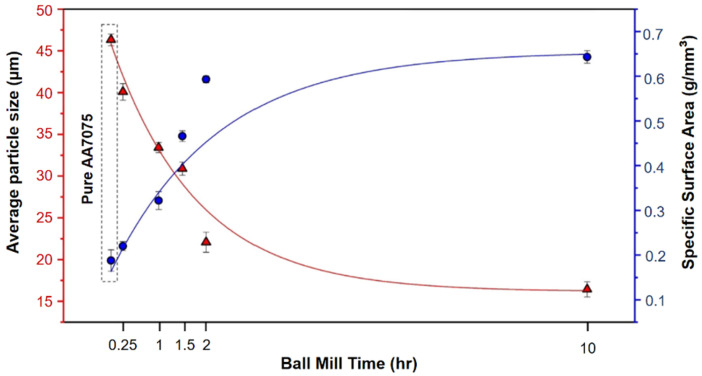
The changes in average particle size and specific surface area of the powders in relation to milling time [55].

**Figure 7 materials-16-04635-f007:**
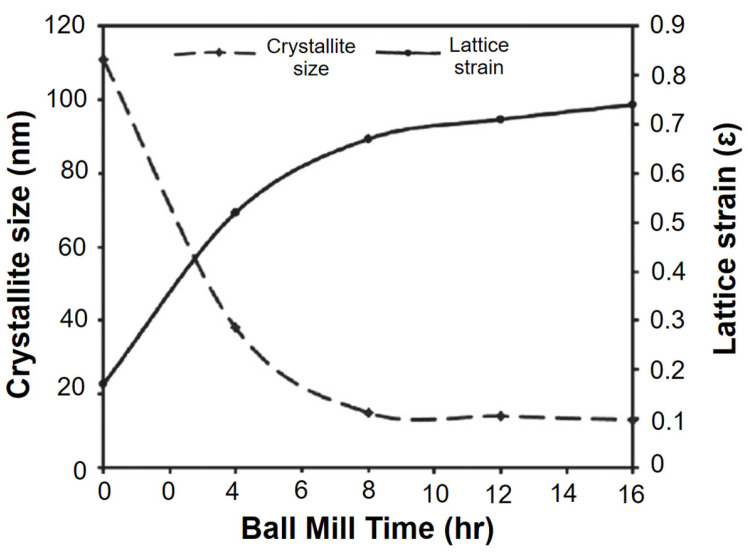
The variations in crystallite size and lattice strain of ZnO powder particles as a function of milling times [123].

**Figure 8 materials-16-04635-f008:**
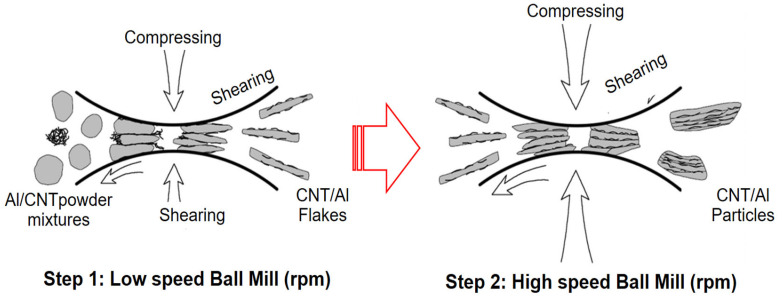
The co-deformation and dispersion mechanisms of CNT/Al powders [126].

**Figure 9 materials-16-04635-f009:**
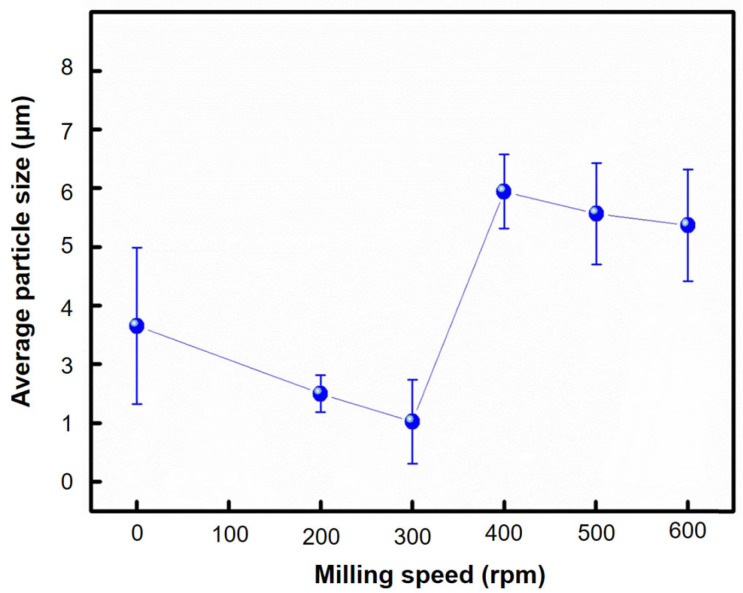
The average particle size of the Ni_70_Mn_30_ powder as a function of different ball milling speeds [129].

**Figure 10 materials-16-04635-f010:**
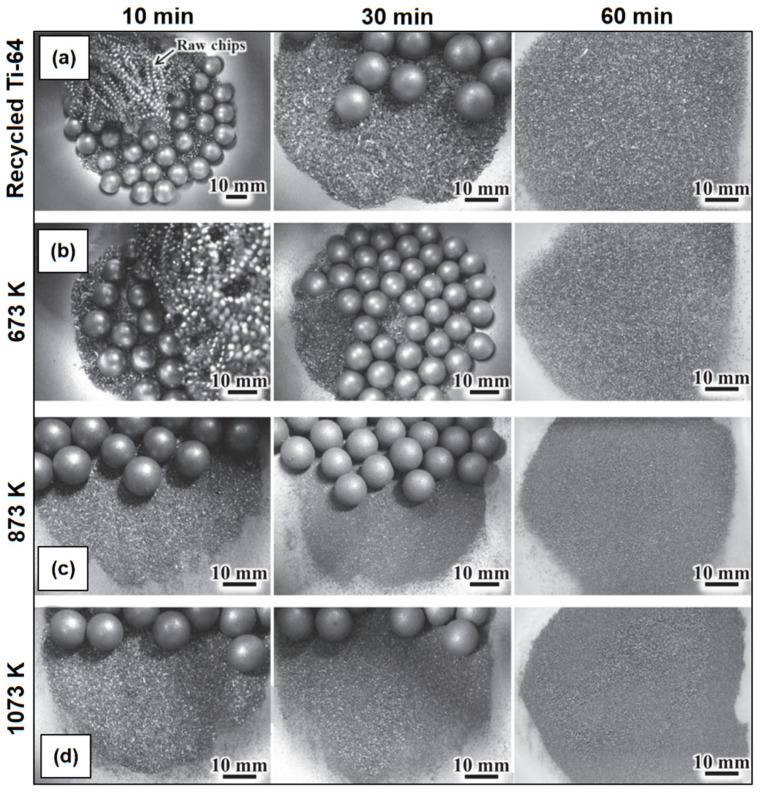
The morphological changes of Ti-64 chips after three different ball milling times: (**a**) raw chips; chips heat treated at (**b**) 673 K, (**c**) 873 K, and (**d**) 1073 K [138].

**Figure 11 materials-16-04635-f011:**
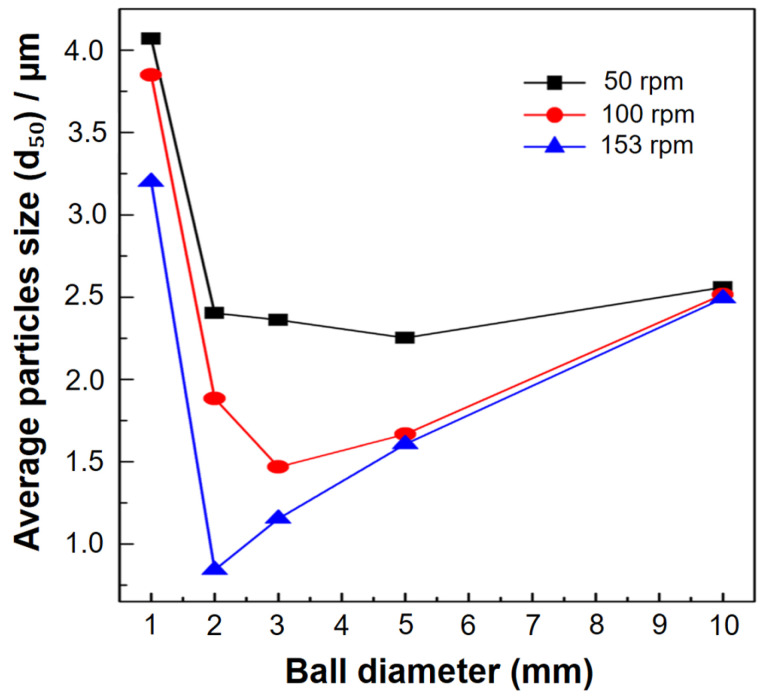
The average particle size (d_50_) of milled Al_2_O_3_ powder with respect to the rotation velocity (rpm) and ball diameter [150].

**Figure 12 materials-16-04635-f012:**
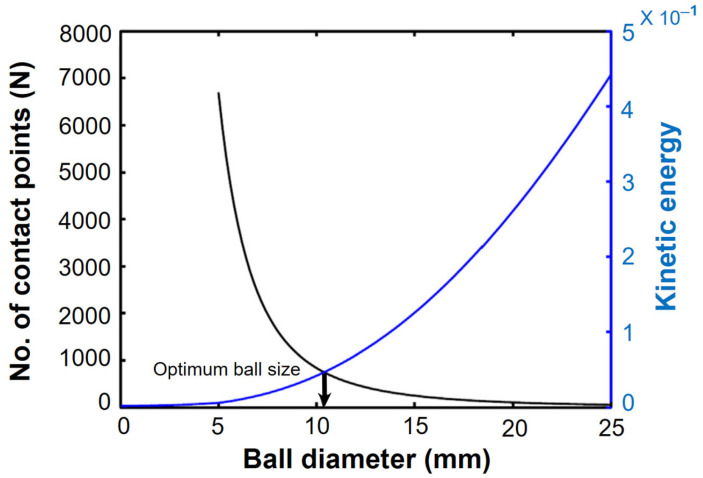
The optimization of ball diameter [156].

**Figure 13 materials-16-04635-f013:**
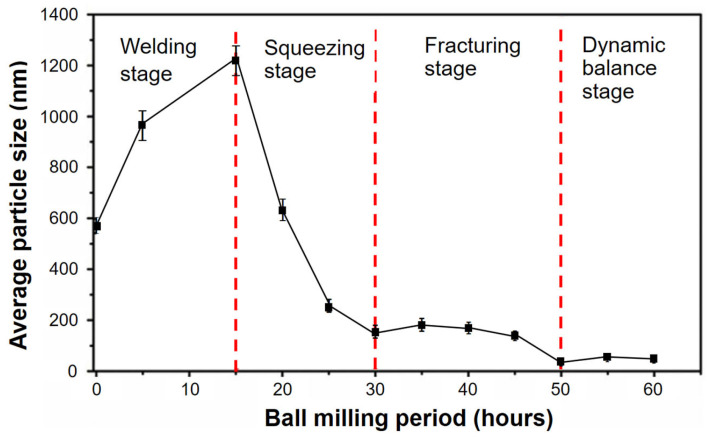
Average of tungsten particles and different ball milling periods [77].

**Figure 14 materials-16-04635-f014:**
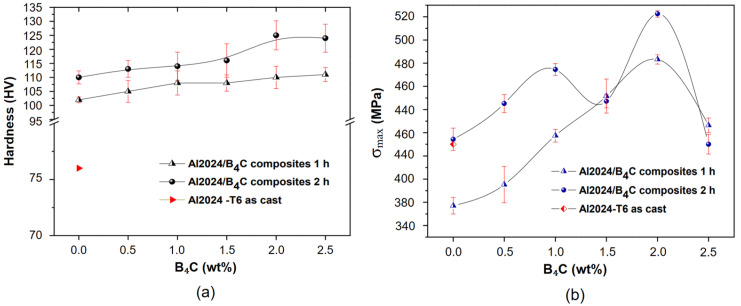
Impact of the B_4_C content and milling duration on the (**a**) microhardness and (**b**) maximum strength of the Al2024/B_4_C composites [78].

**Figure 15 materials-16-04635-f015:**
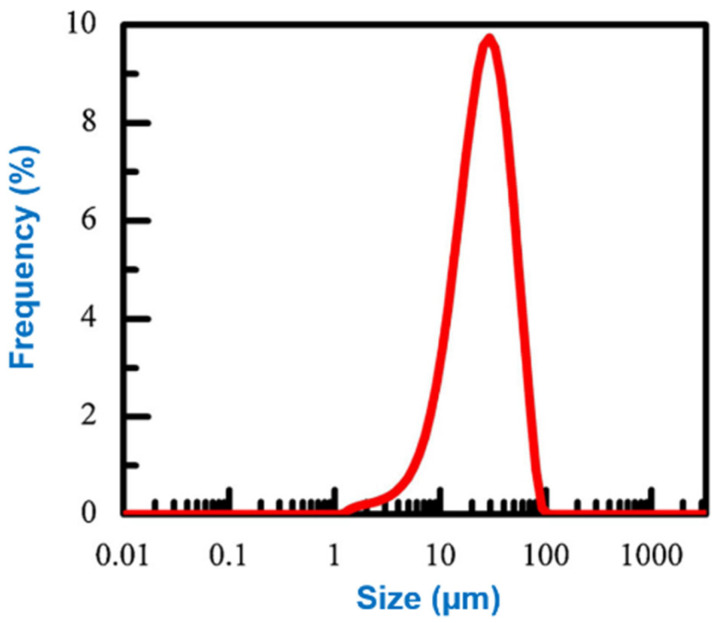
The raw Ni powder’s particle size distribution [79].

**Figure 16 materials-16-04635-f016:**
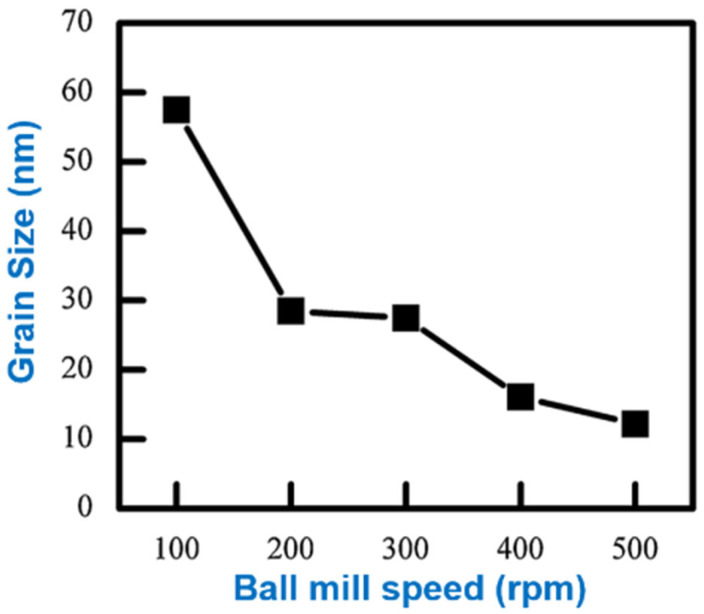
The grain size of milled samples with varying rotation speeds [79].

**Figure 17 materials-16-04635-f017:**
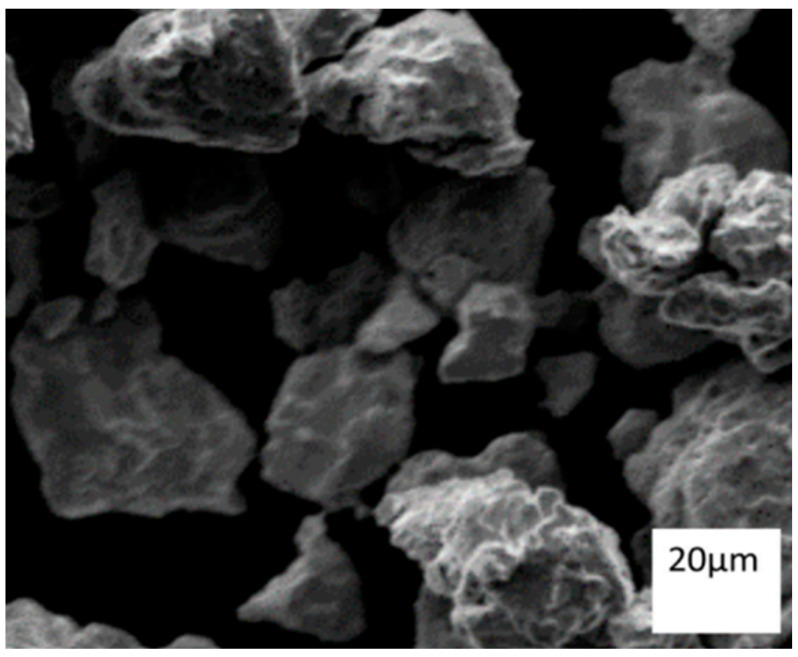
SEM micrographs of stainless steel powder with grinding settings of rotation speed 350 rpm, mass/ball ratio of 1/20, and grinding duration of 50 h [168].

**Figure 18 materials-16-04635-f018:**
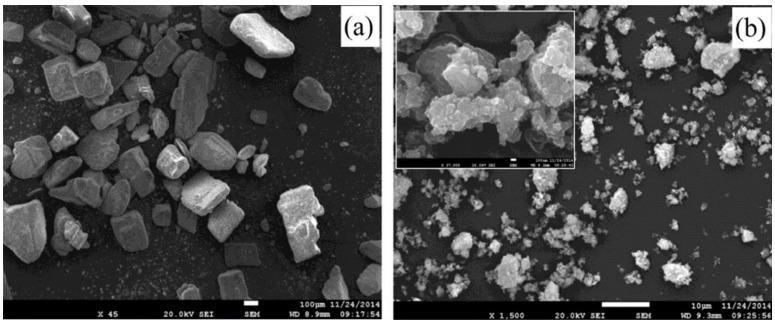
SEM morphology of the specimen (**a**) before and (**b**) after 3 h ball milling [191].

**Figure 19 materials-16-04635-f019:**
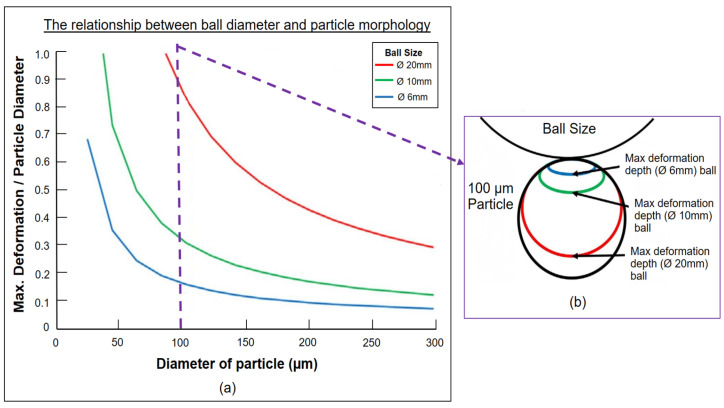
(**a**) Impacted particle normalized maximum deformation depth modeling results by Ø 20 mm, Ø 10 mm, and Ø 6 mm grinding balls as a function of the particle diameter. (**b**) The greatest deformation depth caused on a 100 μm particle by the collision of Ø 20 mm (red line), Ø 10 mm (green line), and Ø 6 mm (blue line) balls are depicted schematically [148].

**Table 1 materials-16-04635-t001:** Ball milling media selection based on a number of desirable characteristics [161].

Materials	Density (g/cm^3^)	Vickers Hardness (GPa)	Appropriate for Usage with	Abrasion Resistance
Agate, SiO_2_	2.65	12.6	Soft to medium hard samples	Good
Corundum, Al_2_O_3_	3.8	20	Medium-hard, fibrous samples	Fairly good
Silicon nitride, Si_3_N_4_	3.25	17.5	Abrasive samples	Excellent
Zirconium oxide, ZrO_2_	5.9	12.5	Abrasive, fibrous samples	Very good
Stainless steel (Fe, Cr, Ni)	7.8	2.4	Medium-hard, brittle samples	Fairly good
Tempered steel (Fe, Cr)	7.9	3.29–3.99	Hard, brittle samples	Good
Tungsten carbide Composites (WC/Co)	14.9	15	Hard, abrasive samples	Very good

**Table 2 materials-16-04635-t002:** Research on the effects of different ball mill jars on mechanical properties.

No.	Researchers	Material	Ball Mill Jar	Input Parameters	Mechanical Properties	Findings
1	Enayati et al. (2007) [69]	Stainless steel scrap chips (average length of 2–4 mm)	Planetary ball mill Hardened chromium steel: Bowl (120 mL), 5 pcs Ball (Ø 20 mm)	Milling speed (rpm): 400. Milling time (h): 25, 50, 100. BPR: 10:1. Cleaned by acetone before ball mill.	Hardness	After 50 and 100 h of ball milling, the mean particle size was 300 μm and 60 μm, respectively. After 50h (820 Hv) and 100h (850 Hv). After annealing, 50h (630 Hv) and 100 (710 Hv). After ball milling, the hardness value is better than metal chips and higher than without the annealing procedure.
2	Liang et al. (2017) [77]	Tungsten (W) powder (purity 99.9%) Particle size: (0.57 µm)	Fritsch P-7 planetary ball mill Zirconium oxide (ZrO_2_): Vials and balls	Speed (rpm): 500. Weight ratio of ball to powder: 15:1.	N/A	After 5h, majority size (4–1.2 μm). After 20h, the majority were smaller than 0.5 μm. After 45 h, 90% between 0.05 and 0.25 μm. After 50 h, particle size remains consistently below 0.1 µm without any reduction.
3	Gallardo et al. (2018) [78]	Al2024 alloy swarf B_4_C (Ø 7 µm) Mixed with different concentrations (0.0, 0.5, 1.0, 1.5, 2.0, 2.5 wt.%)	Dual High-Energy Ball Mills (8000D Mixer/Mills) High chromium steel (AISI D2): Vial and balls	Milling time (h): 1 and 2. BPR: 5:1. Argon atmosphere protection. Cold pressed (900 MPa) for 3 min. Steel die: Billets with (Ø 6 mm), height (12 mm). Sintering: 3 h (15–500 °C). Artificially aged (T6 temper): 6 h at 191 °C.	Micro hardness Compressive strength	After 1 h, the particles have a uniform particle size distribution and an equiaxed shape. Increasing the amount of B_4_C in the Al2024 matrix had no effect on crystallite size, even when milled for 2 h. Best properties, hardness (125 HV), yield point (440 MPa), and maximum strength (520 MPa) were obtained with 2.0 wt% B_4_C at 2 h milling.
4.	Sarah et al. (2019) [103]	Aluminum Swarf (±70 g) with 3 mm after blended	Planetary Ball Mill (Fritsch P-6 Planetary Monomill) Zirconia (ZrO_2_): Container 250 mL, Ball (10 mm x 36 pcs)	Speed (rpm): 370. Run: 3 min. Pause: 27 min. 1 Operation cycle set: 5 run/pause cycles. Acetone (2 mL/g of Al) with ultrasonic waves 1 h. Sieved with the 600 Mesh test sieve (0.025 mm) after milling.	N/A	34.829 µm at least 7 h of milling to homogenously pulverize aluminum. No employ gas atomization emits no harmful gases into the atmosphere.
5.	Li (2020) [79]	Commercially pure Ni powder with 325 mesh (more than 99.8% purity, Alfa Aesar)	High-energy ball milling by QM-3SP2 planetary ball mill WC: Jar (100 mL), Ball (Ø 6 mm)	Milling speed (rpm): 100, 200, 300, 400, 500. Time (h): 1 h milling and 1 h rest. BPR: 15:1. Run: 16 cycles. Milling Environments: argon gas, three types of air are normal, dry, and normal with ethanol.	N/A	After milling in air, the grain and particle sizes are less than 0.02 and 2 μm, respectively.

**Table 3 materials-16-04635-t003:** Research on the effects of different ratio of ball mill on mechanical properties.

No.	Researchers	Material	Ball Mill Jar	Input Parameters	Ratio of Ball Mill Jar	Findings
1.	Biyik and Aydin (2014) [63]	Cu powders (44 µm) Tungsten powders (12 µm)	Fritsch Pulverisette 6 Planetary Ball Mill Tungsten Carbide (WC): Container 225 mL, Ball (Ø 10 mm)	Speed (rpm): 300. Milling Time (h): 0.5, 1, 4, 7, 10, 13, 16, 20, 25. Pause: 30 min. Milling Atmosphere: Air. Chamber Temperature (°C): Room Temperature.	1:10	The optimum amount of stearic acid for the copper-tungsten system was determined to be 2 wt.% by weight of the final particle size (0.726 microns) after 25 h of milling. Microhardness values rose (240 HV), with the highest value obtained in powders (stearic acid content of 2%) after milling for 25 h.
2.	Guaglianoni et al. (2015) [80]	WC-12wt% Co	Pulverisette 6 planetary mill. Bowl: Tempered steel Ball: AISI 52,100 steel (Ø 10 mm)	Milling speed (rpm): 250, 500. Milling time (h): 1, 5, 10.	1:5 1:20	At a BPR of 1:20, a rotation speed of 500 rpm, and milling times of 5 and 10 h, the smallest crystallite size is 0.0138 and 0.0137 µm, respectively. The average particle size (D50) and crystallite sizes were 1.63 µm and 0.0138 µm, respectively, providing a surface area of 4.709 m2/g. (Using BPR 1:20 and milling speed 500 rpm with 5 h).
3.	Petrovic et al. (2018) [183]	Titanium dioxide (TiO_2_)—Cerium dioxide (CeO_2_) powders	Fritsch planetary ball mill Silicon nitride: Vual (80 cm^3^), Ball (Ø 10 mm)	Speed (rpm): 150–400. Milling time (min): 15–141. Powder for each run: 4.87 g. TiO_2_:CeO_2_ weight percentage ratio: (90:10)- (40:60). Samples were calcined for 2 h at 500 °C.	1:10	The band gap energy of TiO_2_:CeO_2_ nanoparticles decreases with increasing milling speed and milling time. The weight percentage ratio of TiO_2_:CeO_2_ and milling speed is more significant. TiO_2_:CeO_2_ weight % ratio (71:29), milling speed (200 rpm), and milling duration had the highest photocatalytic activity (115 min). Higher milling speed promotes anatase-to-rutile transformation.
4.	Vasamsetti et al. (2018) [181]	Rice Husk Ash (RHA)	Planetary ball mill Tungsten carbide (WC): Jar (internal Ø 79 mm) and ball 7.7 gm (Ø 10 mm)	Speed (rpm): 250, 475, 500. Milling time (h): 10, 20, 30.	1:5 1:10 1:15	Ball to powder ratio of 10:1 is enough to obtain better results. The particle size of the resulting powder is less impacted by the size of the balls. Milling speed and duration are the two most essential elements influencing particle size. After 10 h of grinding at 500 rpm and a ball-to-powder ratio of 15:1, a particle size of 0.056 µm was obtained.
5.	Mendonca et al. (2019) [168]	UNS S31803 stainless steel scrap Average size: 8 mm Add with and without vanadium carbide (VC)	High-energy planetary ball mill	Speed (rpm): 250, 350. Milling time (h): 10, 50. Carbide percentage (%): 0, 3.	1:10 1:20	The carbide has the greatest influence, followed by the grinding time, the rotation speed, and the powder-to-ball weight ratio. At 350 rpm, 50 h of milling time, a mass/ball ratio of 1:20, and the addition of 3% VC, particles with sizes from 20 to 135 µm were detected.

**Table 4 materials-16-04635-t004:** Independent variables’ actual and coded values in an experimental design [183].

Variable and Designate	Code Level Ranges and Actual Values				
	−α	−1	0	+1	+α
TiO_2_:CeO_2_ weight percentage ratio, X_1_, (%)	90:10	80:20	63:35	50:50	40:60
Milling speed, X_2_, (rpm)	149	200	275	350	400
Milling speed, X_3_,(min)	15	40	77.5	115	141

Central composite design’s axial point, α = 1.682.

**Table 5 materials-16-04635-t005:** Research on the effect of different PCA types on the ball milling process.

No.	Researchers	Materials	Ball Mill Jar	Input Parameters	PCA	Findings
1.	Zhang et al. (2017) [191]	Multi-layer ceramic capacitors and waste PCB (Cu 57.11% and Zn 15.55%)	Planetary Ball Mill Corundum pot 250 mL (Ø 100 mm) Zirconia ball 120 g (Ø 9 mm)	Speed (rpm): 300 to 600. Mixed in 0.5 mol/L diluted. Hydrochloric acid (HCl) after milling. Stirrer: 15 min. Filtrated by a suction filter, the residue was collected and dried in an oven at 105 °C.	Six groups of co-milling reagents were used to mill (NaCl, NH4Cl, EDTA.2Na, and the mixture of K2S2O8 (a strong solid oxidant) and each of the above three reagents.	Laser Particle Size Analyzers: 0.23 µm after ball milling. Cu increased slightly when the rotation speed was increased from 300 to 400 rpm and then remained unchanged. 600 rpm was the appropriate rotation speed for the MC reaction. Do not use nitric acid or a mixture of nitric acid and sulfuric acid or hydrochloric acid. Therefore, avoid gaseous pollution.
2.	Fullenwider et al. (2019) [148]	304 L stainless steel (length 5–20 mm)	PQ-N04 planetary ball mill Stainless steel: Jar (inner Ø 52 mm), Ball Ø (20 mm and 6 mm)	Milling speed (rpm): 500. Milling time (h): Ball Ø 20 mm (24, 60), Ball Ø 6 mm (12, 24, 36, 60) BPR: 15:1. Ball mill 5 min (on and off) cycles to prevent milling medium overheating. Milling media filled: (approximately 50–60% of jar volume).	Argon environment	At 12 h, the ball mill was stopped, and powder samples were gathered to examine the morphological progression. The balls (Ø 20 mm) reduce the machining chips to a coarse powder. Then balls (Ø 6 mm) formed spherical morphology in the powders with particle sizes ranging from 38 μm to 150 μm (60 h). The ball-milled powder produced from machining chips has a 56% greater hardness than the gas-atomized powder. The hardness was increased by 44% in the ball mill powder particles compared to that of machining chips.
3.	Prosviryakov at al. (2018) [47]	Copper chips (1 mm) Soda–lime silicate glass (50 mm) constant 40%	Retsch PM400 planetary ball mill	Milling speed: 300 rpm. Milling time (h): 1 to 7. BPR: 4:1.	Argon atmosphere	The distribution of nonmetallic particles in copper is inhomogeneous after 1 h of milling. After 7 h of milling, the particle size is less than 100 μm. Milling time increases microhardness by about 320 HV, dislocation density, and copper grain size.
4.	Bhouri and Mzali (2019) [192]	2017 Aluminum alloy chips (length between 20 and 75 mm, width under 1 mm, thickness 0.25 mm)	Planetary ball mill High-power tungsten carbide mixing ball mill. (40 g of broken chips)	Milling speed: 250 rpm. Milling time (h): 1, 2.5, 5, 10, 20. BPR: 10:1.	Methanol (3 mL)	After 2.5 h and up to 20 h, there may be a slight reduction in particle size. About 90 μm decrease in particle size at the conclusion of 20 h. Particle size affects sintered density after hot compaction under 150 MPa.
5.	Dias et al. (2021) [105]	Aluminum bronze chips without lubricants (10 mm), vanadium carbide (VC) powder, and stearic acid powder	Planetary ball mill Stainless steel vial 600 g of balls with the same proportion of diameters: 21, 13, and 8 mm.	Milling speed (rpm): 250, 300, 350. Milling time (h): 10, 30, 50. BPR: 20:1. VC (%): 1, 2, 3.	Inert argon atmosphere. 30 g of chips were weighed and blended with 1% stearic acid	A milling speed of 350 rpm, 3% VC, and a milling time of 50 h all helped to speed up the size reduction of the particles, as shown in DOE tests 8 and 16 with values of 6.57 μm and 7.15 μm, respectively. As a result of VC’s increased hardness, ductile-brittle interactions occur between particles.

## Data Availability

Not applicable.

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
