# Peer review of "Producing Metal Powder from Machining Chips Using Ball Milling Process: A Review"

_materials, 2023, doi:10.3390/ma16134635_

Round 1

Reviewer 1 Report

This article reviews producing metal powder from machining chips using ball milling process. Given the importance of the subject, it could be considered for publication after the following corrections:

·         Please highlight the major findings and conclusions in the field. Improve the abstract, future trends and conclusions sections. Please highlight your review methodology as well.

·         The authors are encouraged to highlight a discussion section and critically discuss more the existing literature.

·         Figures 3 and 4 are nearly duplicated, the authors can refer to Figure 3 when needed and remove Figure 4.

·         The work appears to have been carried out with care, but the authors should highlight their own findings and discussions that are concluded from the review.

Author Response

Thanks for your comments, we really appreciate it. Please refer to the attached file for detailed responses to the comments.

Reviewer 2 Report

1. In the abstract, the third sentence wrote, This method.... However, the above context did not mention which method is referring to, so in this sentence, the term This method is very confusing. 

2. The author needs to strengthen the logical coherence of each research. Instead of analyzing each research individually, it would be better if the author could strengthen the description of the relationships between various studies.

1. The authors should carefully review the language throughout the entire text as there are some errors. For example, in the fourth paragraph of section 2.1, the authors wrote, ‘Wang et al. [102], The researchers investigated... Such errors should be fixed. 

Author Response

(The authors gave the same response as above.)

Reviewer 3 Report

I want to congratulate the authors for the manuscript titled as “Producing Metal Powder from Machining Chips Using Ball Milling Process: A Review”. The review article is interesting and deserves the attention of readers. However, there are several points in the article that require further explanation.

It is important to show the novelty of this review paper in the abstract. Also, please give one-two sentence/s information about the key findings of your paper, mostly about the practical significance. What is the place of ball milling/mechanical alloying (MA) in industrial applications? Is it useful, reliable? What is the success rate of these systems? Do authors recommend it for the future studies? etc.

After analyzing the literature, show before formulating the goal of the "blank" spots. Which has not been previously done by other researchers? You already show the importance of the research being undertaken. But show what will be the new research approach in this review article. You need to show a hypothesis. In the last paragraph of the introduction, add scientific novelty and practical relevance. Add a clear purpose to the article. Considering the introduction part, it can be said that it has been written beautifully but need to include recent published papers on different metal based produced samples (Al, Cu, Mg, Fe, etc.) and ball milling and reinforcement effect on the overall performance of these groups such as:

In the introduction section or other sections, the following papers can be shown for the good comparison of different production process:

https://doi.org/10.1007/s43452-022-00374-z https://doi.org/10.1016/j.measurement.2022.111715 https://doi.org/10.1016/j.apt.2021.08.031

For the section 2, ball milling variables and their process-performance relationship must be explained in detail, including the different types of reinforcement types, usage areas, final product properties (mechanical, tribological), types, benefits or disadvantages of ball milling process on the overall properties of produced materials, etc. The information given is quite superficial. This section is the heart of the article due to the main topic is relevant. Therefore, ı highly recommend the revise that section.

For the other sections, they need to be extended with more literature papers or some subtopics (such as production process, microstructural and crystallographic properties and resultant mechanical properties) can be added. Some of these section include few citations. In a review paper, more and more examples have to be presented. Please take seriously all the citation warnings for perfectly present this paper. Besides, each section must be widened more tables regarding its sections. Also, total figures are very limited for such a review article, please add more figures in each sections.

Considering that the paper is about relationship between ball milling process and resultant material properties, I propose to extend the topic in the mechanical properties section. In addition to that when the authors mention about the mechanical properties it is inevitable to explain the place of the hardening/strengthening mechanisms especially for Orowan, work hardening, etc. mechanisms. Therefore, it is highly recommended to add several references about this subject.

In conclusion section, Challenges and prospects are not enough. At least 4-5 items have to be placed for each. The written items are not useful as well. It is recommend that they should be revised. The authors need to add the advantageous sides of the ball milling process and other related parameters for the practical applications.

Language used in the manuscript is generally satisfying. However, writers should pay more attention of singular / plural nouns. Also, they should control the spell check/ punctuation of words and sentences. Please check all manuscript for language and misspellings. Also, please recheck upper and lower case letter. In addition, spaces should be added between words and numbers. The authors can use suitable grammar-checking software / use the help of a native English speaker to correct these mistakes. Please fix the typographical and eventual language problems in paper. Such as in table 1 materials should be written in subscript.

The layout of the tables should be checked and corrected. Some texts in the figures are written in blue and some in black. It would be better if it was written in a single type. 734. line should be pulled to 733. There is a slippage in the description of Figure 16 and should be corrected. This is review article so would be better off with nomenclature and a graphical abstract.

english needs minor modifications

Author Response

(The authors gave the same response as above.)

Reviewer 4 Report

The lack of primary raw materials is becoming more and more noticeable. Their resources are becoming scarce and the degree of contamination until recently caused a lack of interest in their acquisition and further processing. Using complex in terms of composition and increasingly polluted secondary raw materials, including waste serums, it is necessary to look for such ways of processing these raw materials that the entire process of their recovery would be profitable both in economic and environmental terms.

In this context, the paper  - Producing Metal Powder from Machining Chips Using Ball Milling Process: A Review - takes on special significance. The authors undertook to review works on the recycling of metal waste from industry and workshops. The paper is valuable. However, it has significant shortcomings.

Main disadvantages:

Section „Introduction”

The grinding process in the ball mill is devoted to Section 2. Therefore, the content concerning this type of grinding should be redrafted and moved to Section 2.

In addition:

Caption under figure 1 - remove the ] sign at the end of the caption.

Caption under Figure 2 - put a full stop at the end of the caption.

First paragraph after figure 3. - Second sentence with reference to literature [54] - Repeat from previous paragraph - reword.

The last sentence of the next paragraph " There are several advantages to recycling metal waste::" should be moved to a new paragraph.

In the following paragraph of the sentence” Additionally, metal recycling is essential for resource conservation, energy conservation, waste reduction, and economic benefits. Copper particles are a potential material for high-quality composite materials due to their conductivity, stability, and low manufacturing cost. Future research should focus on making the process of recycling copper particles and other metals more efficient.” – are a repetition; remove recommendation.

In the next paragraph – after the reference [81], remove the space.

Next paragraph - Content repeated in previous paragraphs - reword, do not repeat the same content.

Next paragraph - What is a planetary process - the term does not appear earlier. Reword - in the context of the content of section 2.

- Section 2

Section 2.1. Important Processing Parameters in the Ball Mill Process

Last paragraph. - Similar content - either redraft or delete.

Section 2.1.2. Milling time - Reconcile content with previous information; do not repeat the same information.

Section 2.1.3. Milling Speed - Reconcile content with prior information; do not repeat the same information.

Section 2.1.4. Type of milling container - Agree content with previous information; do not repeat the same information.

Subchapter 2.1.5. Milling atmosphere - Agree content with previous information; do not repeat the same information.

Section 2.1.6. Milling temperature - Reconcile content with previous information; do not repeat the same information.

Section 2.1.7. Ball size distribution - The part of sentence " It is identified by the mass fraction of balls of various sizes " - information commonly known - delete. The last paragraph of the subchapter - Reconcile the content with the previous information; do not repeat the same information.

Section 2.2. Selecting ball mill media

The last paragraph - Reconcile the content with previous information; do not repeat the same information.

Section 2.2.1. - Ball Mill Jar and Balls - Reconcile content with previous information; do not repeat the same information. What is the content of point 2.2. differ from the content of point 2.1.

Section 2.2.2. - Alternative Materials for Ball Mill Jar and Balls The last paragraph - repeated content - delete.

Section 2.3. Ball Mill Reconcile content with previous information; do not repeat the same information.

Section 2.3.1. Impact of Different Ball Mill Jar on Mechanical Properties First paragraph - Reconcile content with previous information; do not repeat the same information.

Section 2.3.2. Effect of Different Ratio of Ball Mill on Mechanical Properties first paragraph; second and third sentences - repeated information - delete.

Caption under Figure 11 - remove the space after literature [166].

Section 2.4.3. Effect of Different ethanol of ball mill process. – Can 2.3.3. ?

Part of the last paragraph “table 7 [32, 60, 92, 177–178]” combine with the whole paragraph.

Author Response

(The authors gave the same response as above.)

Round 2

Reviewer 2 Report

The author has made revisions according to the reviewer's comments, and it is recommended to accept.

Reviewer 4 Report

I accept the authors' comments